# Mechanical Design and Performance Analyses of a Rubber-Based Peristaltic Micro-Dosing Pump

**Thomas Zehetbauer [1,\*], Andreas Plöckinger [1], Carina Emminger [2]⬤ and Umut D. Çakmak [2,\*]⬤**

[1] Linz Center of Mechatronics GmbH (LCM), Altenbergerstrasse 69, 4040 Linz, Austria; andreas.ploeckinger@lcm.at

[2] Institute of Polymer Product Engineering, Johannes Kepler University Linz, Altenbergerstrasse 69, 4040 Linz, Austria; carina.emminger@jku.at

\* Correspondence: thomas.zehetbauer@lcm.at (T.Z.); umut.cakmak@jku.at (U.D.Ç.)

**Abstract:** Low pressure fluid transport (1) applications often require low and precise volumetric flow rates (2) including low leakage to reduce additional costly and complex sensors. A peristaltic pump design (3) was realized, with the fluid's flexible transport channel formed by a solid cavity and a wobbling plate comprising a rigid and a soft layer (4). In operation, the wobbling plate is driven externally by an electric motor, hence, the soft layer is contracted and unloaded (5) during pump-cycles transporting fluid from low to high pressure sides. A thorough characterization of the pump system is required to design and dimension the components of the peristaltic pump. To capture all these parameters and their dependencies on various operation-states, often complex and long-lasting dynamic 3D FE-simulations are required. We present, here, a holistic design methodology (6) including analytical as well as numerical calculations, and experimental validations for a peristaltic pump with certain specifications of flow-rate range, maximum pressures, and temperatures. An experimental material selection process is established and material data of candidate materials (7) (liquid silicone rubber, acrylonitrile rubber, thermoplastic-elastomer) are directly applied to predict the required drive torque. For the prediction, a semi-physical, analytical model was derived and validated by characterizing the pump prototype.

**Keywords:** hydraulic pump; micro-dosing; peristaltic; hyper-elasticity; viscoelasticity; holistic design methodology; elastomer compound

## 1. Introduction

Pumps have a broad field of application and can be considered as energy transducers, converting primary kinetic energy (e.g., linear, or rotational motion of a rigid body) to hydrodynamic energy [1]. A simplified view on the technical side of a pump reduces it into three main components which describe the operation principle [hydraulic pump very abstracted]: housing with fluid in- and outlet port, moving component(s) and transmission gear to drive the moved part via the primary energy source. Based on the operation principle hydraulic pumps, beside some exceptions like "the hydraulic ram" [2] which uses the water hammer effect [3] as primary energy source, can be categorized into two main groups, namely centrifugal pumps [4] and (positive) displacement pumps. Schmitz and Murrenhoff [5] gives a good overview of hydraulics in general. Centrifugal pumps have an open fluid connection from in- to outlet port, the impeller accelerates the fluid due to its rotational movement which causes centripetal forces (actio); in other words: the fluid is moved due to its centrifugal force caused by the impeller (reactio). In contrast to that, the in- and outlet ports of displacement pumps are disconnected by a sealing which is considered as "leak-free" flow, so the fluid volume "the displacement" is encapsulated and transported by the motion per turn. Some pumps of this kind have multi-sectioned and even parallel, and phase shifted displacement to smooth the flow rate and, consequently, reduce pulsations. The most common types of displacement pumps are gear pumps, screw

pumps, rotary vane pumps, as well as piston pumps, which can further be divided into axial and radial piston pumps [6,7] and, finally, peristaltic pumps [8]. Except for peristaltic pumps, these displacement pumps have several features in common. The dynamic sealings, which enclose the displacement, are usually not intended to be made of soft materials and, therefore, consist of long and tight sealing gaps. Consequently, an acceptable amount of leakage occurs. Lower leakage requires more precise manufacturing and is expensive. The fact that such pumps mainly consist of rigid components with significantly higher strength than soft materials, higher maximum pressures ($p > 21$ MPa) can be withstood compared to peristaltic pumps.

Other research activities pursue the application of smart (soft) electroactive materials [9,10] in peristaltic pumps. Furthermore, the combination of (micro)pumps with flow sensors is shown by Jenke et al. [11] and others to ensure closed loop controlled micro-dosing. However, the degree of complexity is increasing by the implementation of the additional sensors.

Our aim was to build a cost-effective and backflow-free (safety valve function) fluid transport system which can handle pressure ranges from nearly 0 MPa to 1 MPa with accurate but variable flow rates. Furthermore, it must withstand the dry operation mode and also negative pressures (relative to atmospheric pressure). For these specifications, the most appropriate system is the peristaltic pump. However, some unfavourable issues like low durability or pulsation are well-known [8] and must be addressed in the development of peristaltic micro-dosing pumps.

Figure 1 shows the main principle components of a peristaltic pump with three rollers including a cross-section of it. The main problems include pulsations caused by discontinuity of displacement per turn as well as durability limitations and dynamic thermomechanical behaviour of the wobbling plate's soft material [12–14], among others. So, the problems can be related to the pump concept itself and the undesirable behaviour [15] (strength, aging, creep, relaxation, etc.) of the soft material. On the one hand the benefit of the hyper-elastic behaviour is the conformity, flexibility, and the tight dynamic sealing capability [16]. On the other hand, this soft (elastomeric) material will be adiabatically heated by the pump motion and accelerate the material's aging. Standard tubes of common peristaltic pumps are pushed by one or more rotating rollers to encapsulate the displacement (fluid) into one or more segments (Figure 1). Depending on the design, such squeezing leads to high deformations and unbalanced contact-pressures. Due to zones of high inner stresses, especially on kinks (i.e., highly deformed contact regions), the material will embrittle over time and, ultimately, leading to total failure of the pump. To overcome these, in this article the main objective was to present a new mechanical design approach of a rubber-based fluid pump including prototyping and validation.

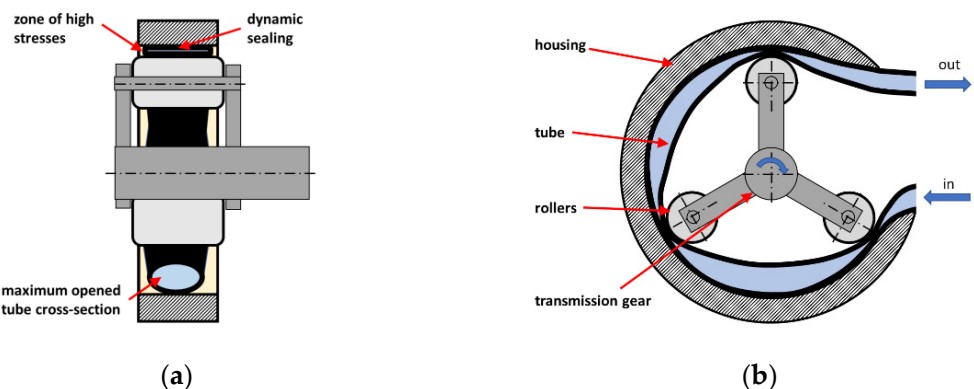

(a)                                                                                                    (b)

**Figure 1.** Flexible tube based peristaltic pump principle with 3 rollers; (**a**) cross-section of the pump in frontal-view; (**b**) cross-section of the pump in side-view.

## 2. Design Concept and Challenges

Figure 2 shows the workflow of the design methodology. The methodology comprises the mechanical design of the peristaltic micro-dosing pump, the material selection approach of the wobbling plate's soft layer and the optimization of the whole pump system. It is an interwoven approach including analytical as well as numerical calculations and experimental validations.

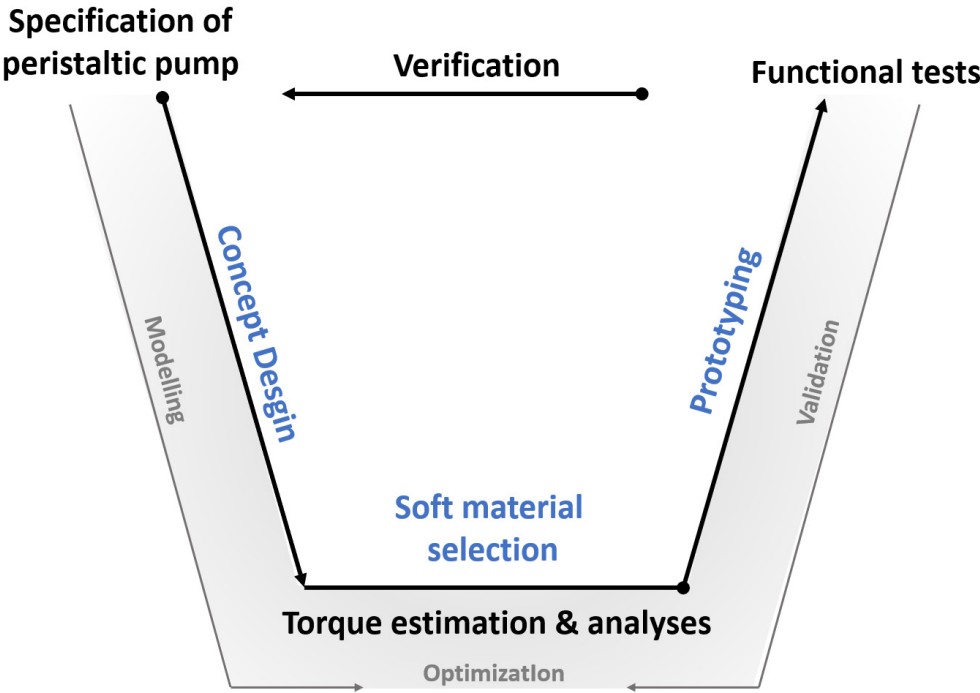

**Figure 2.** V-model of the peristaltic micro-dosing pump.

With our methodology, long-lasting FE-calculations to compute a functioning pump geometry at the maximum drive conditions (torque) are avoided to analyse leak-free fluid transport at pressure levels from 0 MPa to 1 MPa. Furthermore, it enables operation at variable flow rates (μL/min to mL/min) as well as service-temperatures (0–60 °C).

### 2.1. Mechanical Design

Figure 3 shows the mechanical design and a prototype of the peristaltic micro-dosing pump system. The pump concept has a mostly constant displacement, enclosed by the pump cavity and the moving dynamic wobbling plate, which transports the fluid through the provided channel. The flow rate can be varied by setting the wobble turn speed appropriately. As primary pump systems drive, we chose a slightly oversized electronic commutated brushless direct current motor [17] due to the off-the-shelf availability and the good controllability.

The design of the pump system was a result of an easy way to adjust the dynamic sealing pressure by altering the distance of the two subassemblies, namely the pump and drive part.

For the mechanical design of the peristaltic pumps' primary drive, the estimation of the drive's torque in relation to the established sealing pressure within the cavity is of particular importance. Here, the wobbling plate is made of a rigid and a soft layer (rubber) and, hence, the sealing pressure is mainly determined by the material behaviour of the soft layer.

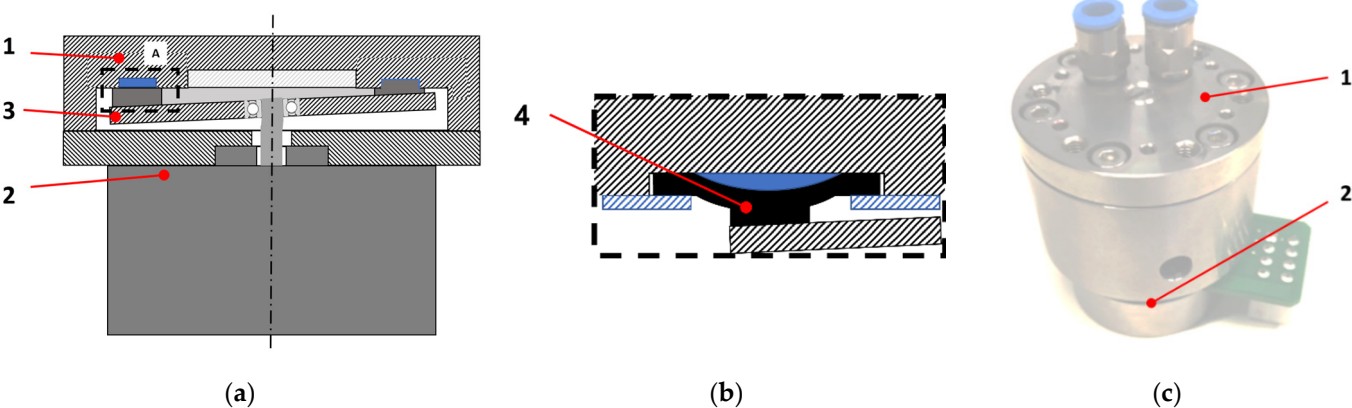

**Figure 3.** Design of the micro-dosing pump, pump itself (1), primary drive with transmission gear (2) and the non-rotating wobble plate (3) with the soft material layer on top which forms the cavity together with the housing; (**a**) Cross-section view of the schematic illustration; (**b**) detailed cross-section of the rubber (4) and cavity; (**c**) picture of the prototype.

### 2.1.1. FE-Calculations and Design Hypothesis

For the geometric design, Abaqus [18] was used to calculate material stresses and deformations. The main observed operating points are those in extreme conditions, which must be tolerated by the materials. The general pump design was simulated as a 2D axis-symmetric model. The steps were primarily set as dynamic-implicit (quasi-static) without fluid dynamics but with pressures on the sealing boundary zones. In the first step, the mounting process was simulated. Therefore, the basic pump housing was fixed, and the other parts ($y$-axisymmetric) were moveable in y-direction and fixed in x-direction. The prestress-rings were now brought into its final positions. In the further Steps the boundary conditions were applied (rising with step time) from zero to its nominal values. On each step, one of the boundary conditions was changed. Its limits were maximum and minimum pressures (minimum 0 MPa, nominal 1 MPa in operating condition, and maximum pressure of 3 MPa in none-operating pump state for safety reasons) and wobble angles (which results in a maximum of 0.5 mm and minimum of 0 mm in y-deformation of the rubber)

The soft material was modelled with isotropic and hyper-elastic via Mooney–Rivlin constitutive behaviour at certain temperatures and frequencies. For faster and more stable simulation results the step-types were set as dynamic, implicit with adiabatic heating effects. Due to large deformation of the rubber, the nonlinear geometric function is enabled. Several mesh sizes and elements were tested in order to find a stable model. In particular, the rubber element is the crucial component and therefore needs a more specific consideration. The seeds very applied evenly spaced around the circumference of the 2D axisymmetric rubber element. Whereas the minimum occurring cross-section of the rubber has a size of $5 \times 2$ mm$^2$ the approximate global size was set to 0.12 mm and curvature control was used for applying the global seeds. The maximum deviation factor was set to 0.1 as set by default in ABAQUS CAE. In mesh controls the element shape was set to be "Quad" only. Also "Free" technique with "Advanced front" algorithm included "mapped meshing where appropriate" was chosen. Furthermore, the mesh element type was set as "CAX8RH" (An 8-node biquadratic axisymmetric quadrilateral, hybrid, linear pressure, reduced integration). Now the rubber part can be meshed automatically, and as seen in Figure 4b the mesh size and structure are evenly distributed, and the shape of the rubber element is sufficiently accurate.

There contact formulations between the parts was separated into three segments. All part interactions were set to "all with itself" with cohesive contact ("Hard" contact for normal behaviour and default cohesive behaviour) excluding those surfaces which are not permanently attached to each other. The surface contacts between the rubber and the pump head, and between pump head and prestress ring (in Figure 4 Position 2 and 1,

Position 1 and 3) were chosen to be of type surface-to-surface interaction. The rubber surface and the prestress ring surface were set to be the slave-surfaces and the pump head was set as master surface. The contact behaviour was formulated as slip-friction. Therefore, the tangential behaviour was set es penalty friction formulation with a coefficient of 0.15 and isotropic directionality. Additionally, the normal contact behaviour was set as "Hard" contact for pressure-overclosure with default constraint enforcement method. Separation after contact was allowed to simulate other situations in further "Steps". After the first "Step" was created, the initial boundary conditions were applied. Further "Steps" were implemented with the appropriate change of boundary conditions and loads.

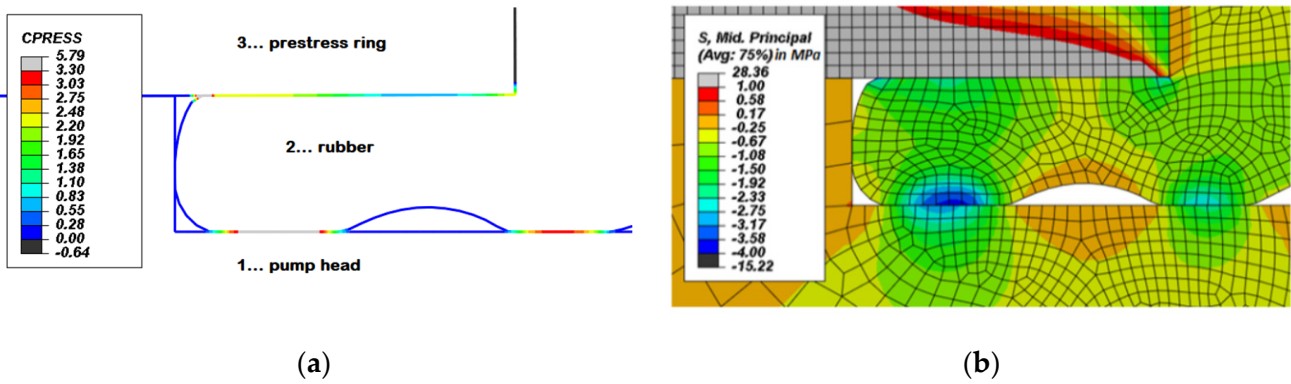

(**a**)  (**b**)

**Figure 4.** First results of 2D axisymmetric FE-Calculation: (**a**) resulting contact pressure of the static sealing lips after assembling; (**b**) resulting stresses of the rubber due to the prestress. Due to reasons of presentability the geometry was flipped in contrast to detail A of Figure 3.

The basic design parameters of the cavity geometry were chosen due to desired flow rate (50 mL/min), which is the product of displacement (33 µL + 20% over-dimensioned for geometry tolerances) and motor speed (1500 rpm). From this point on, the 2D-FE-calculation procedure was iterated to find a basic pump design concept working for the given boundary conditions to fulfil all desired conditions like displacement and maximum pressures without violating the maximum allowable stresses. The maximum contact pressures for "Step 1" on the sealing lips (see Figure 4) are above the desired maximum static pressure. It can be concluded that there will be no external leakage on the sealing lips, also the maximum allowed stresses were not exceeded. All steps results fulfilled the desired conditions without violating the maximum allowed material conditions like maximum deformation and maximum tensile and compressive stresses.

The 2D calculation led to the final design parameters like rubber thickness, maximum deformation for prestressing the sealing lips and maximum compression at maximum wobble angle which separates high-pressure and low-pressure area. After the basic design was finalized, a 3D-FE-analysis was conducted for one single pump revolution at one specific operating point specified as room temperature and quasi-static movement. This special situation represented very slow pump motions and, therefore, low flow rates. Of course, such a simulation is very time-consuming (several weeks of run-time). As a result, we received the verification of the functionality of the basic concept (closed displacement in every rotary angle) and a first scale of the necessary torque to move the wobble plate properly. The necessary drive torque is mainly influenced by the rubber conditions like deformation frequency and temperature, but also by the required hydraulic power which can later be superimposed easily. To have a whole dynamic Abaqus 3D-design calculation, it would be necessary to simulate several operating points like maximum frequency at minimum temperature with the same pre-set assembling conditions, which has been omitted due to long simulation times.

### 2.1.2. Analytical Model for the Estimation for the Drive Torque

An analytical model estimating the necessary drive torque for several operating points (different speeds and temperatures) is derived to avoid expensive dynamic 3D-FE-analyses by using the results of the viscoelastic characterization of the wobble plate's soft layer by dynamic thermal mechanical analyses (DMTA). For the model, some level of abstraction is needed to describe the function as realistic as possible without too many deviations. So, the first step is finding a simplified scheme of the pump system including the primary drive and power transmission which represents the rubber (soft layer) dominated required torque (idle mode) in a sufficiently accurate manner as seen in Figure 5a,b.

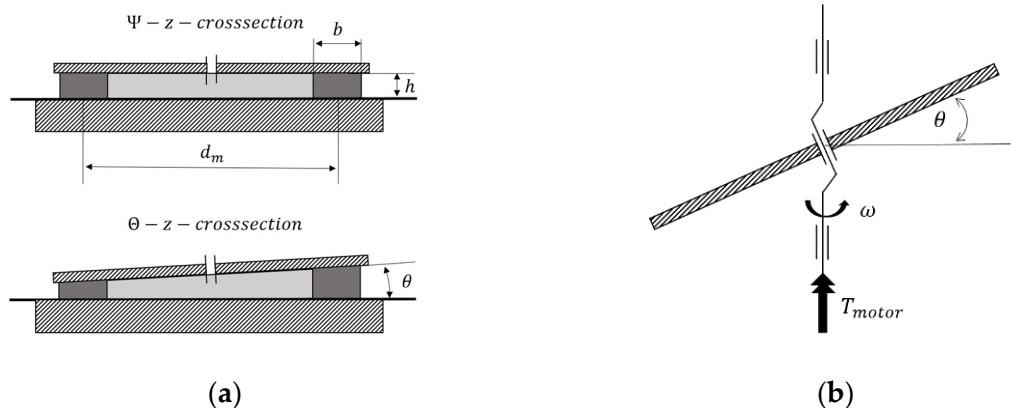

(**a**)          (**b**)

**Figure 5.** Simplified abstracted model of the pump system: (**a**) wobble plate on top connected with the rubber ring in zero and maximum deflection; (**b**) illustration of the transmission from rotary drive to wobbling plate (the roller bearing allows a rotation between wobble plate and drive shaft, the wobble plate itself is blocked due to the attached and fixed rubber).

The mechanics of the wobbling plate according to Figure 6 can be formulated with the equation of motion for angular problems with 2 degrees of freedom $\vec{q} = \begin{pmatrix} \alpha & \beta \end{pmatrix}^T$ and looks like:

$$J \cdot \ddot{\vec{q}} + D \cdot \dot{\vec{q}} + C \cdot \vec{q} = \vec{T}, \tag{1}$$

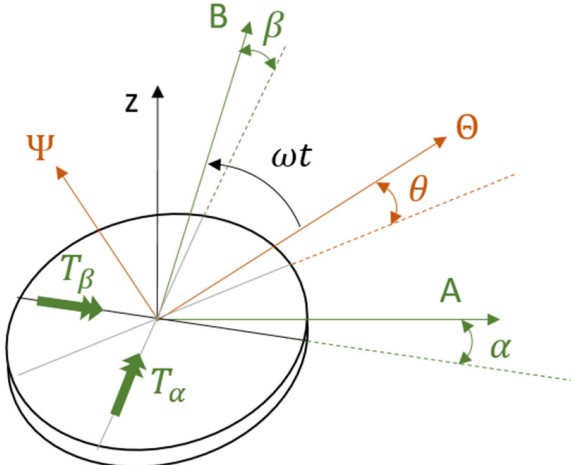

**Figure 6.** Wobbling plate with labelled kinematic parameters and torques.

Considering a harmonic movement of the plate, where $\theta$ changes with $\varphi = \omega t$ which leads to a harmonic wobbling, the state vector can be written as follows:

$$\vec{q} = \begin{pmatrix} \alpha \\ \beta \end{pmatrix} = \begin{pmatrix} \theta \cdot \sin(\omega t) \\ \theta \cdot \cos(\omega t) \end{pmatrix}, \tag{2}$$

The first and second derivations lead to:

$$\dot{\vec{q}} = \begin{pmatrix} \dot{\alpha} \\ \dot{\beta} \end{pmatrix} = \begin{pmatrix} \theta \cdot \omega \cdot cos(\omega t) \\ -\theta \cdot \omega \cdot sin(\omega t) \end{pmatrix}, \tag{3}$$

and

$$\ddot{\vec{q}} = \begin{pmatrix} \ddot{\alpha} \\ \ddot{\beta} \end{pmatrix} = \begin{pmatrix} -\theta \cdot \omega^2 \cdot \sin(\omega t) \\ -\theta \cdot \omega^2 \cdot \cos(\omega t) \end{pmatrix}, \tag{4}$$

Furthermore, the needed power for each rotation is constant, therefore, it is possible to observe the resulting values at a certain quasi-static position (e.g., $\varphi = \omega t = 2\pi k \ \forall k \in \mathbb{N}$), so the state vector and its first and second derivations simplify to:

$$\vec{q} = \begin{pmatrix} 0 \\ \theta \end{pmatrix}, \tag{5}$$

$$\dot{\vec{q}} = \begin{pmatrix} \dot{\alpha} \\ \dot{\beta} \end{pmatrix} = \begin{pmatrix} \theta \cdot \omega \\ 0 \end{pmatrix}, \tag{6}$$

$$\ddot{\vec{q}} = \begin{pmatrix} \ddot{\alpha} \\ \ddot{\beta} \end{pmatrix} = \begin{pmatrix} 0 \\ -\theta \cdot \omega^2 \end{pmatrix}, \tag{7}$$

The ideal power transmission from the rotary drive to the wobbling as illustrated in Figure 5b results in following formulation:

$$T_{motor} \cdot \omega = T_\alpha \cdot \dot{\alpha} + T_\beta \cdot \dot{\beta} \tag{8}$$

Considering the simplifications of the states in Equations (5)–(7) due to quasi-static and position-based observation, Equation (8) further simplifies to:

$$T_{motor} = T_\alpha \cdot \theta \tag{9}$$

These results substituted by the original terms of equation of motion (1) lead to the following time invariant algebraic equation:

$$T_{motor} = d_\theta \cdot \theta^2 \cdot \omega \tag{10}$$

The damping and stiffness coefficients for this problem "rubber ring with tilting load" according to Figure 5a can be written as followed [19]:

$$c_{\theta(T,\omega)} = \frac{E'(T,\omega) \cdot \pi d_m^2 \cdot \frac{b}{h}}{8} \left( \frac{1 + \left(\frac{b}{d_m}\right)^2}{1 - v^2} + \frac{b^2}{3h^2} \right), \tag{11}$$

and

$$d_{\theta(T,\omega)} = c_{\theta(T,\omega)} \frac{\tan(\delta)}{\omega}, \tag{12}$$

Merging Equations (10)–(12) the required torque for idle mode can be estimated by the following equation:

$$T_{motor} = \frac{E'(T,\omega) \cdot \pi d_m^2 \cdot \frac{b}{h}}{8} \left( \frac{1 + \left(\frac{b}{d_m}\right)^2}{1 - v^2} + \frac{b^2}{3h^2} \right) \cdot \theta^2 \cdot \tan(\delta), \tag{13}$$

With these considerations, an algebraic equation of the very abstracted model of the pump system was found to estimate the required torque which is influenced by temperature

and frequency-depending material behaviour (modulus $E'$ and loss factor tan $(\delta)$) and the geometry parameters. The experimental set up for the validation can be seen in a following sub-chapter called evaluation process. If these torque values are also useable for scalable designs without the need of further investigations will be proven in the results.

### 2.1.3. Dynamic Fluid Gap

Due to the nonlinear dynamic rubber behaviour, the displacement of the maximum pump pressure will be temperature and frequency dependent. Once the rubber's dynamic thermomechanical behaviour is characterized, only operation temperature and pump speed are necessary to estimate the actual flow rate. The requirement for the pump system was a constant displacement independent of speed, temperature and pressure. However, the dynamic fluid gap, which is formed between the squeezed rubber and the rigid pump cavity, has a huge impact on the pump's displacement. For further investigations, the wobble plate is considered as a cubic piece of rubber with a rigid top which is frequently excited and pressed against the pump cavity. The level of abstraction is shown in Figure 7. The distance varies from maximum lift (hmax) to the nominal compression to reach the target contact pressure at minimum lift (hmin).

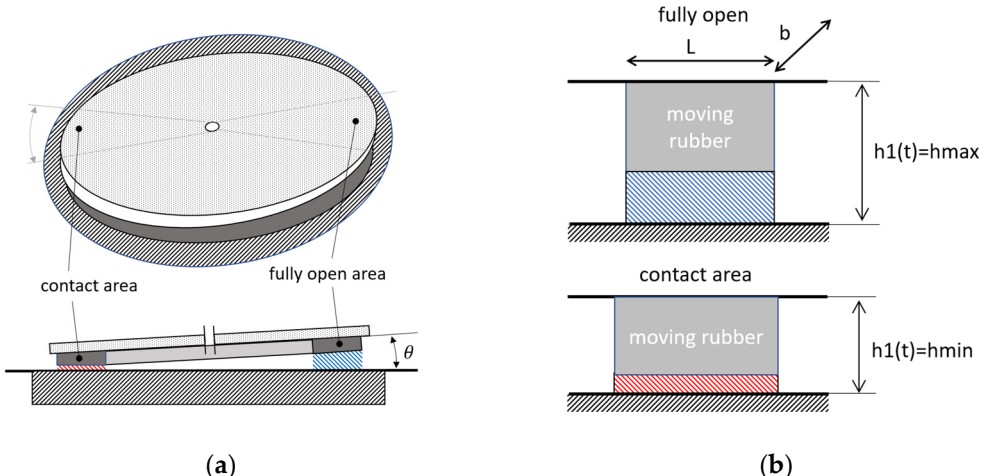

(**a**)           (**b**)

**Figure 7.** Analogous model of the complex pump system regarding the dynamic film on the contact area: (**a**) illustration of the simplified wobble situation; (**b**) principal abstracted scheme.

This dynamic fluid gap is influenced by the stiffness of the rubber, the occurring fluid pressure and the wobbling speed of the pump itself. A phenomenological mechanical model based on this system is illustrated in Figure 8. The rubber was described by the well-known Kelvin–Voigt model [20] (parallel connection of elastic spring and viscous damper).

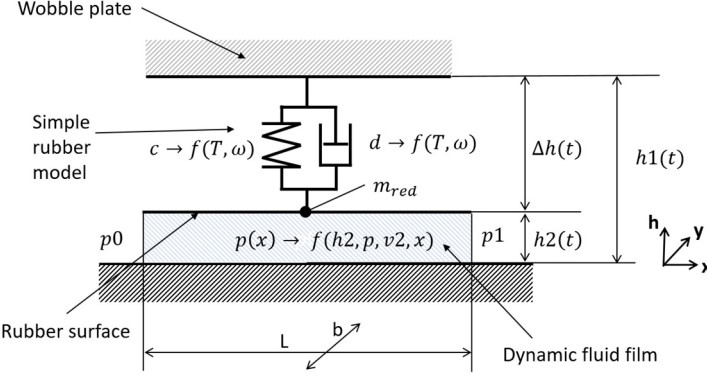

**Figure 8.** The simplified mechanical model of the emerging fluid gap caused by the rubber on the wobble plate.

The Reynolds equation [21,22], which is a certain form of the Navier–Stokes equations [23] (no density or viscosity changes) for a fluid gap with such a short distance leading to dominating viscosity effects, can be written as below:

$$\frac{\partial}{\partial x}\left(\frac{h^3}{\eta}\frac{\partial p}{\partial x}\right) + \frac{\partial}{\partial y}\left(\frac{h^3}{\eta}\frac{\partial p}{\partial y}\right) = 6\frac{\partial h}{\partial x}(U_{12}+U_{21}) + 6\frac{\partial h}{\partial y}(U_{12}+U_{21}) + 12\dot{h}, \quad (14)$$

Furthermore, $y$-axis-depending terms and horizontal velocity-based terms (wobbling plate does not have a relative movement to the cavity in horizontal direction) can be eliminated:

$$\frac{\partial}{\partial x}\left(\frac{h^3}{\eta}\frac{\partial p}{\partial x}\right) = 12\dot{h}, \quad (15)$$

By integrating this differential equation 2 times, and setting the boundary conditions to $p(0) = p0$ and $p(L) = p1$ the damping pressure below the rubber can be calculated:

$$p(x) = -6\eta\frac{\dot{h2}(t)\cdot x^2}{h2(t)^3} + p0 + x\cdot\left(6\eta L\frac{\dot{h2}(t)}{h2(t)^3} + \frac{p0}{L} + \frac{p1}{L}\right), \quad (16)$$

The resulting force of the fluid cushion which hinders the rubber to get into contact with the cavity can be evaluated by integration over $dx$ from 0 to the length $L$ and $dy$ from 0 to the width $b$ of the strip.

$$F_{fluid} = \int_0^b\int_0^L p(x)dydx \rightarrow f\left(h2(t), \dot{h2}(t), p1, p2\right) \quad (17)$$

At an operation point the rubber can be simplified to a frequency and temperature dependent spring and damper system with very low mass. With $\Delta h(t) = h1(t) - h_{rub} - h2(t) - h0$ the equation of motion can be written as follows

$$m\cdot\ddot{h}(t) + d(T,\omega)\Delta\cdot\dot{h}(t) + c(T,\omega)\cdot\Delta h(t) = F_{fluid}\left(h2(t), \dot{h2}(t), p1, p2\right), \quad (18)$$

Damping effects of the rubber cause an internal heating and a lag of deformation, which in this case means, that zero damping would lead to a worst-case scenario for the developing fluid gap.

The mass of the elastomer also has low influence on the results, with $m = d = 0$. The Equation (18) can be reduced as follows:

$$c(T,\omega)\cdot(h1(t) - h_{rub} - h2(t) - h0) = F_{fluid}\left(h2(t), \dot{h2}(t), p1, p2\right), \quad (19)$$

By setting the boundary conditions $p1$ and $p2$ to zero to represent the idle mode the fluid the equation further simplifies to:

$$c(T,\omega)\cdot(h1(t) - h_{rub} - h2(t) - h0) = \frac{L^3 b\eta}{(h2(t) + h0)^3}\cdot\frac{d}{dt}h2(t), \quad (20)$$

To obtain the desired result $h2(t)$, this nonlinear differential equation needs to be solved numerically.

In Figure 9a,b the emerging fluid gap due to different excitation frequencies is shown. The typical rubber behaviour leads to a higher stiffness with rising frequency and falling temperature. Presuming the stiffness rises to infinite values, the fluid gap would only be defined by two rigid plates which lead to zero contact, but infinite high contact forces. On the other side, if the rubber stiffness is very low, the fluid gap will be dominated by the velocity term namely the motor speed respective frequency. In conclusion, depending on the chosen rubber material rising frequency will lead to higher gaps, due to higher fluid forces, but also to an increasing stiffness of the rubber which will partly compensate the

emerging gap. The graphs in Figure 10 give an idea of how the displacement changes due to frequency effects. The nominal flow rate $Q$ is dependent on motor speed $n$ and displacement $V$ which is further a function of $n$.

$$Q = V(n) \cdot n, \tag{21}$$

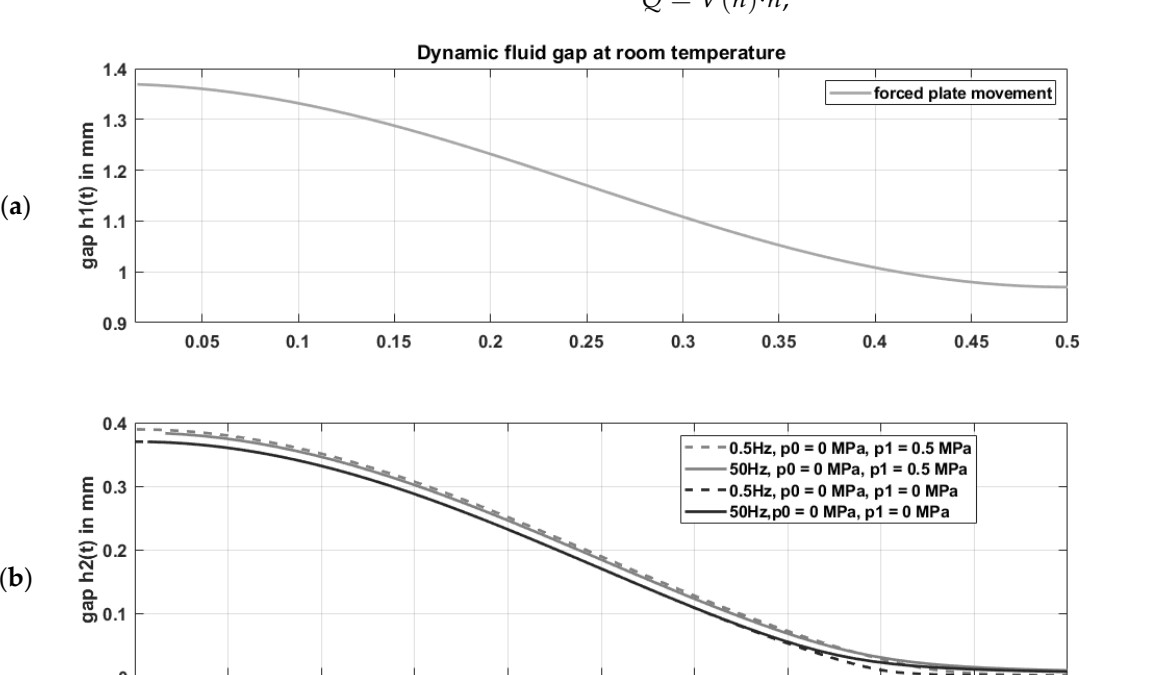

**Figure 9.** First simulation results of the dynamic fluid gap: (**a**): the forced fluid gap between wobble plate and cavity h1(t); (**b**): emerging fluid gap h2(t) due to different motor frequencies and load pressures.

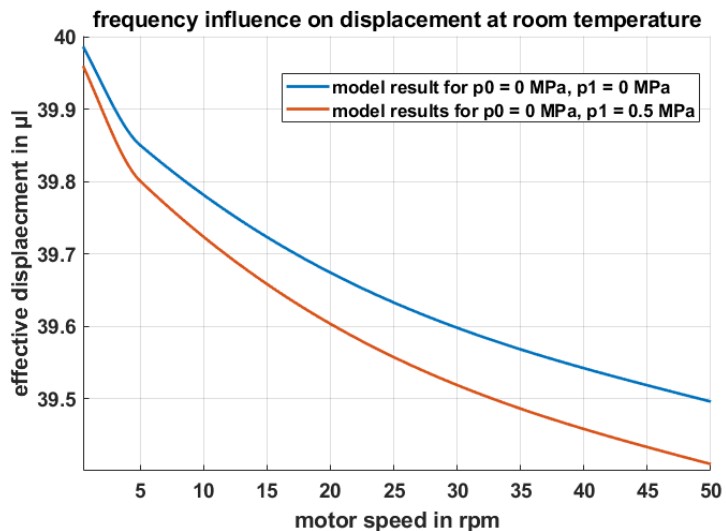

**Figure 10.** Calculated displacement curve based on V0 = 40 μL for idle mode and nominal load pressure over desired motor speed.

The nominal displacement (quasis-static $\rightarrow f = 0$ Hz) is calculated to:

$$V_0 = r_m \oint_0^{2\pi} A(\varphi)d\varphi \approx d_m \cdot \pi \cdot \frac{2}{3} \cdot b \cdot \frac{h2max}{2}, \tag{22}$$

where $d_m$ is the cavity's mean diameter, $A(\varphi)$ the $\varphi$-depending (radial) cross-section which is a about the product of the $\varphi$-depending height $h2(\varphi)$ multiplied by two thirds of the cavity width $b$ (area of circle segment). The integral of the cross-section $A(\varphi)$ results in the mean of maximum and minimum cross-section. The frequency dependent term reduces the nominal displacement to:

$$V(f) \approx V_0 - d_m \cdot \pi \cdot \frac{2}{3} \cdot b \cdot \frac{h2min(f)}{2}, \tag{23}$$

The actual system is much more complex, due to the hyper-elastic as well as viscoelastic material behaviour and the interaction of the more complex pump geometry. Most likely it is not possible to find an analytical model to describe the thermal, dynamic rubber fluid interaction of the 3D design.

Figure 11 shows the pressure distribution from the left to the right boundary in the dynamic fluid gap. The pressure inflation in the dynamic gap at the exact time when the gap has its minimum (sealing point) must be higher than the boundary pressures in order to prevent backflow, since the flow rate follows the negative pressure gradient. Therefore, Figure 11a,b shows that internal leakage is prevented if the boundary pressure is below the maximum pressure at low frequencies. This is valid, if the rubber prestress is high enough, which is guaranteed by the required static sealing contact pressure. From Figure 11c it is obvious that the boundary pressure of 1.5 MPa exceeds its target sealing limit causing internal leakage.

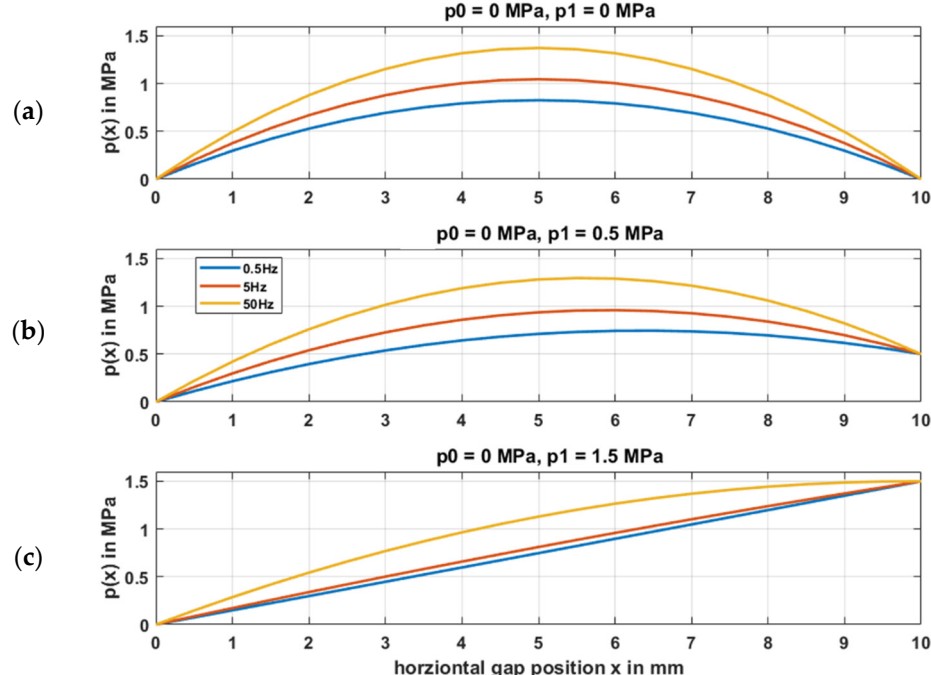

**Figure 11.** Inflated pressure distribution p(x) along the dynamic fluid gap at certain time min(h1(t)): (**a**) boundary conditions p0 = p1 = 0 MPa; (**b**) boundary conditions p0 = 0 MPa; p1 = 0.5 MPa (nominal operating point); (**c**) boundary conditions p0 = 0 MPa; p1 = 1.5 MPa (pressure limit exceeded → backflow from right to left side).

### 2.1.4. Wobble Plate Design

The design of the rubber-based wobble plate has surrounding static sealing lips in a kidney shaped arrangement, which are tight at maximum pressure. Depending on the pivot angle of the wobble plate, the rubber is pressed at the inner side against the pump's housing cavity. The geometry must be designed in order to maintain the contact pressure line just above the desired feed pressure. Otherwise, leak tightness is not preserved. Since

the pump-element also undergoes tensile stresses, the connection between the rigid and the soft layer is crucial and must resist these severe loadings. However, soft materials, like rubbers, exhibit inherent viscoelastic behaviour leading to stress/strain relieving (relaxation/retardation) mechanisms due to the molecular motion under external loading. This effect must be considered in the mechanical design with the aim to reduce it; if not, the displacement of the peristaltic pump will drop and reduce the flow rate permanently.

To overcome the drop in displacement, the boundaries of the rubber must be confined in a way that over time evolving creep along the maximum stress path is inhibited [24]. Therefore, three joining concepts for the soft and the rigid layers were considered in this study and included Figure 12a adhesive bonding with cyanoacrylate Figure 12b vulcanization (cross-linking) of the rubber on the metallic surface by applying a primer for enhanced adhesion between the dissimilar materials, and Figure 12c form fit and vulcanization for superior interface properties.

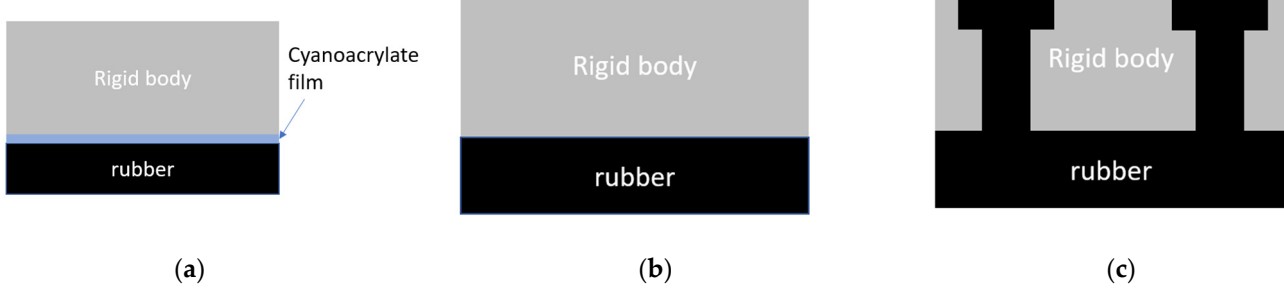

(**a**)    (**b**)    (**c**)

**Figure 12.** Different ways of bonding rubber to a "rigid" body like metal or plastic: (**a**) cyanoacrylate as film; (**b**) vulcanizing makes a tight connection; (**c**) form closure and vulcanizing is the best way to connect rubber with rigid structures (including special surface treatments).

There is a specific dynamic behaviour of the pump due to the nonlinearity of the rubber. Besides the desired pressure, the minimal required torque is mainly influenced by this certain elastomeric behaviour. Therefore, at low frequencies and high temperatures, the torque is mainly given by the hydraulic power (output power of the pump) and the main prestress. Very low temperatures and high frequencies lead to a stiffening effect of the rubber. These operating conditions influence the maximum pressure and the consistency of the flow rate.

*2.2. Material Selection for the Soft Layer of the Wobbling Plate*

In the peristaltic pump design, the wobbling plate squeezes the fluid through the cavity to the outlet port. The soft layer enables the squeezing (i.e., the contraction of the soft layer leading to a pressure in flow direction) and must be flexible, resilient and exhibit low hysteresis under dynamic loading. These design requirements are crucial for a reliable operation of the pump and have to be translated to material properties. The state-of-the-art candidate materials for peristaltic pumps are elastomeric materials (rubbers), crosslinked incompressible polymers, for the general purpose of the functionality to enable squeezing of the fluid. However, this class of materials exhibits non-linear (hyperelastic) material behaviour and has an inherent viscoelasticity (loading rate and temperature dependency). Above the glass-transition temperature, the predominant deformation mechanism of rubbers is governed by changes of entropy. Under deformation the macromolecules are oriented, decreasing the entropy and, thus, leading to adiabatic heating. This mechanism is directly linked to the efficiency of the pump, as the external electrical drive of the pump has to supply higher torques to maintain constant flow rate and pressure. To provide the functionality of the wobbling plate within the above-mentioned constraints and requirements, the objective is to select a material with following properties within the loading frequency and temperature ranges:

- constant storage modulus E′ for low and balanced sealing pressures,
- low loss modulus E″ for low drive torques,
- low Poisson's ratio $\nu$ (rubber is incompressible, hence, $\nu = 0.5$),
- low viscoelasticity and elastic deformability up to 25% strain

Figure 13 illustrates the application-relevant frequency and temperature ranges for the storage and the loss moduli. Including the reversible deformability of up to 25%, these are the most important selection criteria for the material. Focusing on these requirements, the goal is to select (tailor) a material which is sufficiently within the ideal behaviour range of Figure 13.

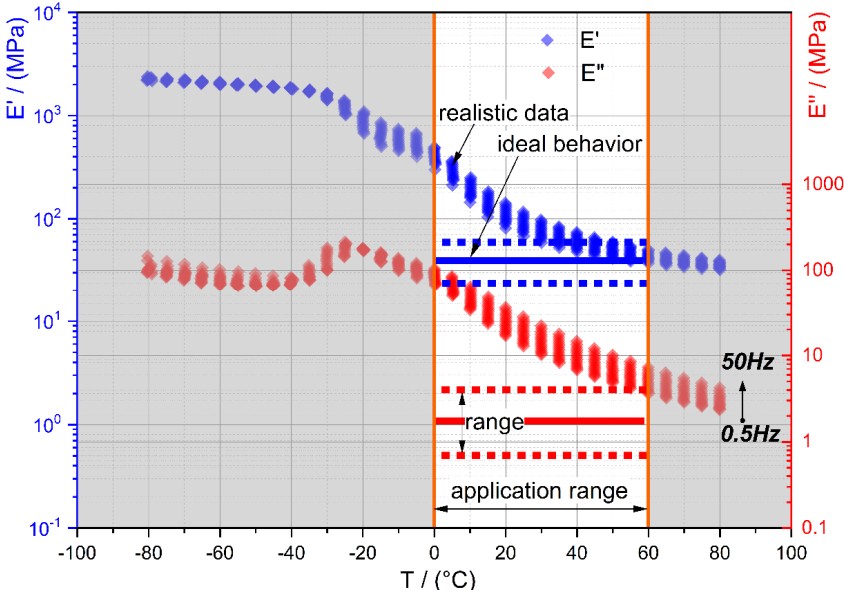

**Figure 13.** Illustration of the realistic material behaviour and the idealistic material behaviour for the soft layer of the wobbling plate.

Based on these considerations, an experimental testing procedure for the material selection is described in the following. As the candidate materials are limited to elastomers (rubber-like materials) along with thermoplastic elastomers, the shear modulus is described by the kinetic theory of rubber elasticity and gives a proportionality of the macroscopic modulus to the molecular mass as well as temperature. The foundation for the material selection procedure is the characterization of the hyper- and viscoelasticity including material parameter determination for numerical simulations to calculate the stress-strain state under application relevant loadings.

A number of hyper-elastic material models are established with their specific limitations of deformation, accuracy and applicability to some reinforced elastomers. The well-known Mooney–Rivlin model is a hyper-elastic model with two material parameters (C01 and C10). This constitutive model is implemented in most commercially available finite element (FE) solvers and describes the material behaviour up to moderate deformations (<30%) and, hence, is applied in our study.

Additionally, the material aging is of particular importance and its effects on the dynamic thermomechanical behaviour of the elastomers have to be examined. For the assessment of the long-term stability of the wobbling plate, the environmental (temperature and humidity) impact on the bonding of the soft layer to the rigid body (cf. Figure 12) as well as the temperature induced aging conditions are crucial. These conditions are addressed experimentally by the cataplasm aging test of the bonding and by thermal aging of the rubbers.

### 2.2.1. Candidate Materials (Rubbers)

From Figure 13 and from previously performed and published works (e.g., [25]), the candidate materials can be narrowed down to rubbers with Shore A hardness of 60 to 80. Specifically, formulations of rubber families of acrylonitrile (NBR, HNBR), fluoro-rubber (FKM), ethylene-propylene-diene (EPDM), thermoplastic polyurethane and silicone rubbers are perfectly suitable as soft layer for wobbling plates in peristaltic pumps. They can accommodate strains of several 100% with low resilience under dynamic loading. The dynamic (thermo-) mechanical properties E′ and E″ are adjustable and the viscoelasticity (frequency and temperature dependency) can be tailored for the application.

In the subsequent chapters we present the experimental data of a liquid silicon rubber (LSR, R401/70), a NBR formulation (Shore A 70) and a TPU formulation (Shore A 90). The LSR and NBR materials were formulated and processed by Erwin Mach Gummitechnik (Hirm, Austria). All three materials can be injection moulded in arbitrary geometries and, therefore, have an economical advantage when high volume mass-production is required.

### 2.2.2. Experimental

The three candidate materials (TPU, LSR and NBR) were moulded to 300 mm × 200 mm sheets with 2 mm thickness. From these sheets the specimens for the hyper- and viscoelastic characterizations were stamped. Figure 14 shows the specimen geometries including the dimensions. On the surface, speckle patterns were coated and during the experiments pictures of the specimens were recorded to derive strain optically by 2D-digital image correlation (2D-DIC; Aramis 4M).

The uni- and biaxial tension tests (see Figure 15) were performed with an electromechanical testing system (TA Instruments, ElectoForce Systems Group) under isothermal conditions at room temperature and at three loading rates (uniaxial: 0.1 mm/s, 1 mm/s, and 10 mm/s; biaxial: 0.1 mm/s, 0.5 mm/s, and 1 mm/s). Strains were derived by 2D-DIC, and forces were measured with a 440 N load cell (WMC-100lbf; Interface Inc., Atlanta, GA, USA). Prior to testing, the specimens were fixed and a dwell time of 5 min was given to reach the thermal equilibrium force (relaxed state of the material).

The material parameter determination was performed by assuming incompressibility and measuring the uni- and biaxial characteristics (compare in Figure 16a range of measured versus application-relevant loading in terms of first and second strain invariant). All experimental data were fitted iteratively to identify the parameters C01 and C10 for the Mooney–Rivlin model (see Figure 16b). The temperature dependency is not captured by this model explicitly. Therefore, dynamic (thermo-) mechanical analyses (DTMA) were performed to gain insights of the temperature and frequency dependencies, on the one hand, and, on the other hand, the storage and loss moduli (loss factor tanδ).

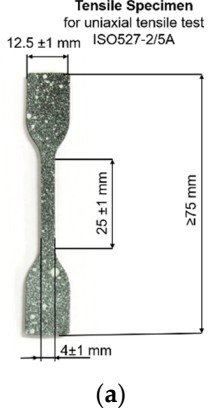
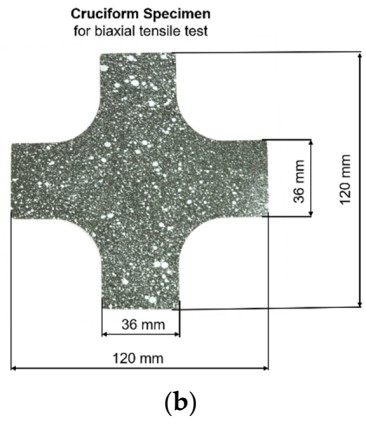

(**a**)  (**b**)

**Figure 14.** Specimens for hyper- and viscoelastic characterizations (**a**) ISO-5A tensile specimen; (**b**) cruciform specimen for in-plane biaxial tension tests. Both specimens were coated with speckle patterns for full field strain analyses.

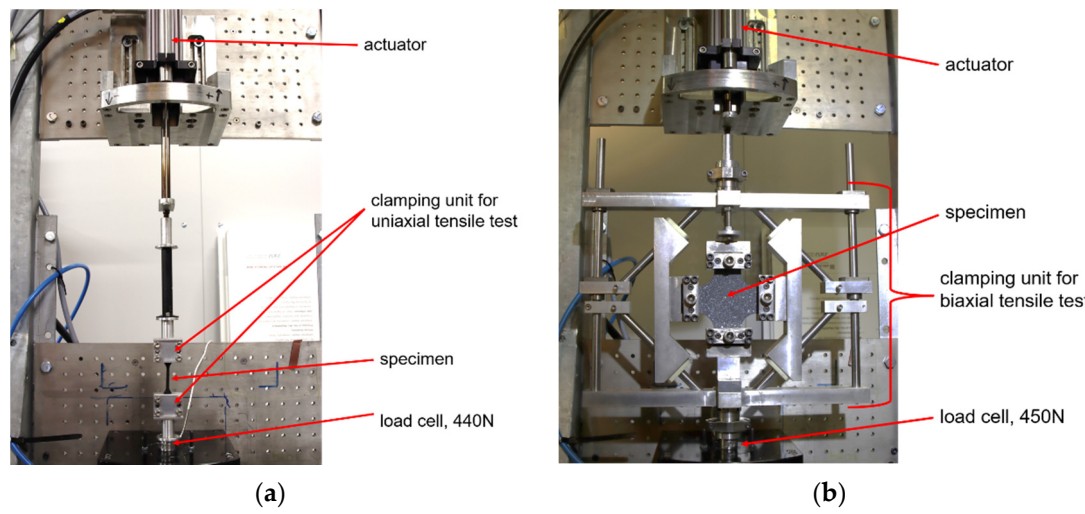

**Figure 15.** Experimental test-setups for: (**a**) uniaxial testing and (**b**) biaxial testing.

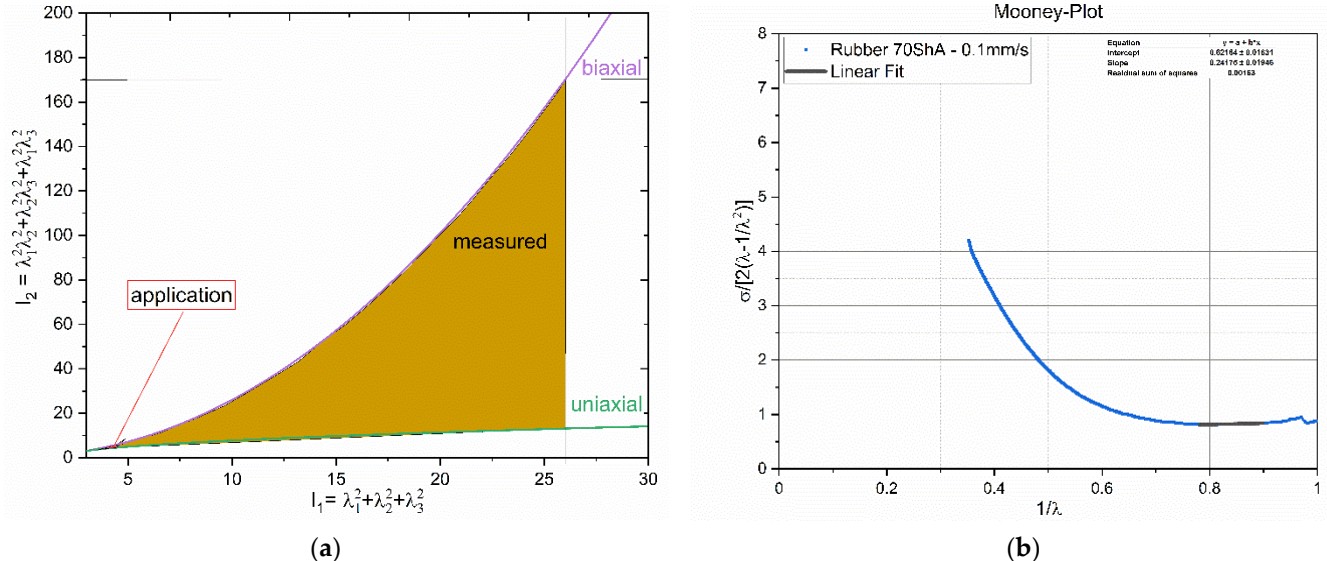

**Figure 16.** Results of the exemplary tests: (**a**) measured strain states (**b**) Mooney-plot and estimated material parameters for the Mooney–Rivlin constitutive model.

The dynamic thermomechanical behaviour of the candidate elastomers was analyzed under uniaxial loading at temperature from $-50$ °C to $+80$ °C and loading frequencies from 0.5 Hz to 50 Hz. A sine wave excitation was applied with a mean strain level of 20% and a dynamic (p-p) amplitude of 2%. DTMA was performed with an Eplexor 500 N (NETZSCH-Gerätebau GmbH, Graz, Austria) and started at the lowest temperatures with an incremental increase of 5 K. The frequency sweep was performed in a logarithmic scale with five frequencies per decade. The obtained material data ($E'$ and $E''$) were further analyzed regarding the thermorheological behaviour and, finally, the master-curves were constructed by shifting and applying time-temperature superposition principle. Figure 17a shows the temperature dependent experimental data of $E'$ and $E''$ at three excitation frequencies (0.5 Hz, 5 Hz, and 50 Hz). The frequency dependent $E'$ master-curve constructed for the reference temperature Tref of 25 °C is illustrated in Figure 17b. For the construction of the master-curve only a horizontal shift was applied. It is important to select an appropriate temperature increment in order to assure an overlap between the isothermal $E'$ curves. Figure 17b shows also the experimental window with the isothermal curves (low to high temperature data are illustrated from top to bottom). Low temperature data are

equivalent to high frequency data (shift to right) and vice versa. As the thermorheological material behaviour was simple, the shift-factors were modelled by the well-known and established Williams–Landel–Ferry equation (WLF).

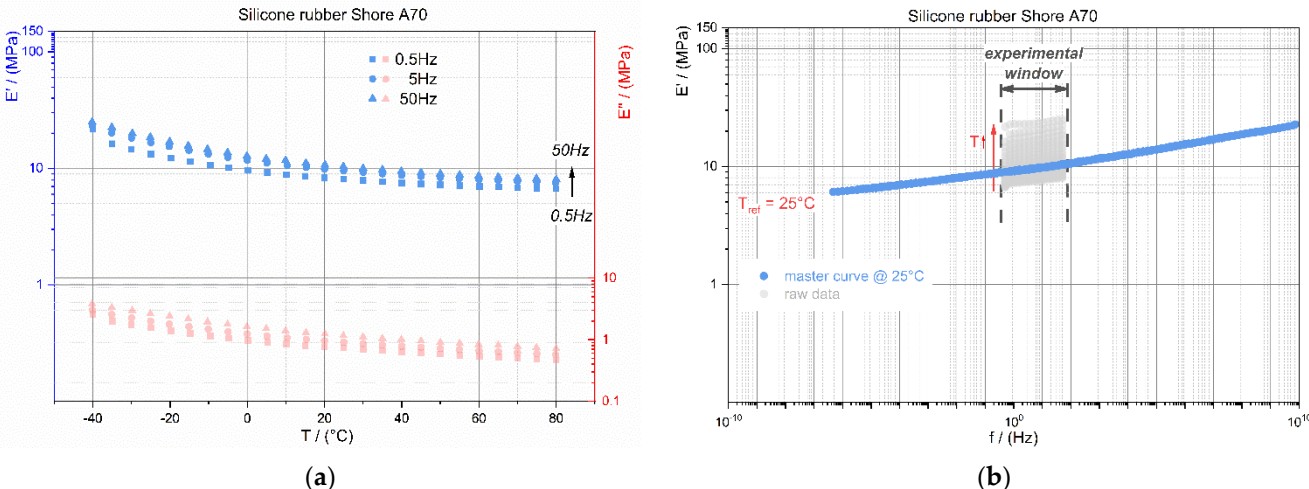

**Figure 17.** DTMA analyses for LSR Shore A 70 (tempered at 150 °C for 2 h); (**a**) loading frequency and temperature dependent storage and loss modulus (E' and E"); (**b**) calculated master curve for a certain nom. temperature.

Material aging alters the properties of the rubbers and, thus, the evolution of the dynamic thermomechanical properties (E' and E") must be characterized. With these insights, the long-term behaviour of the peristaltic pump can be assessed, and maintenance intervals defined. To achieve this, the specimens were exposed to 120 °C for seven days and DTMAs were performed. A precondition was that no surface failures, such as cracks, colour alterations and tackiness [14,26] (sticky touch due to migration of oligomers from the bulk to the surface), were observable.

Another critical aspect to ensure the long-term stability of the pump is the bonding of the wobbling plate's two layers. It is reported [13] that cyanoacrylate is suitable for bonding rubber to other (dissimilar) materials. The chosen super adhesive (Zwaluw Sekundenkleber universal) is a liquid cyanoacrylate-based glue which cures through moisture respective humidity within seconds. The processing temperature should be between 15 °C and 40 °C to reach its final adhesion after half a minute. According to the datasheet, the cured bond withstands a temperature range of −60 °C to 80 °C and maximum mechanical stresses up to 10 MPa (rubber to rubber). Preliminary experiments at room temperature confirmed this, however, the realistic loading conditions of the soft layer are multiaxial at high temperatures and humidity. So, the cataplasm-test [27] was conducted to simulate these severe environmental conditions. Specimens were prepared by cleaning, roughening, and bonding of two rectangular sheets (see Figure 18a) of metal and the candidate materials. After 24 h curing, the specimens were wrapped with water-soaked cotton wool as shown in Figure 18b. In addition, this package was wrapped with aluminium foil, then packed airtight and vacuumed in a PE-bag (hermetic sealed). After 14 days thermal exposure at 70 °C (see Figure 18a), the bonding quality was examined. Bondings passing the cataplasm test without delamination are eligible for the application.

### 2.3. Evaluation Procedure of the Pump Prototype

In the following sections, the procedures for displacement determination, design of the test rig including the control parameters for the performance tests and the evaluation procedure are outlined.

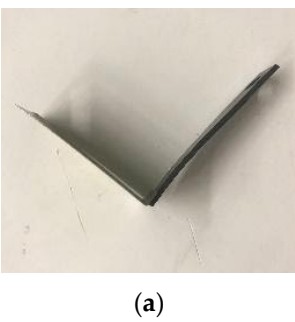
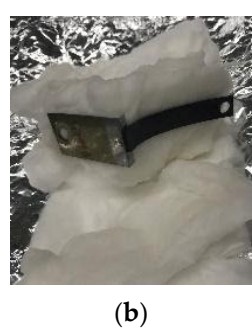
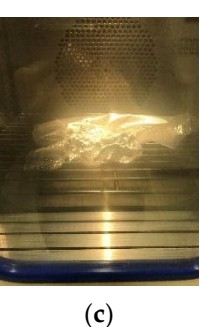

(**a**)　　　　　　　　　　　(**b**)　　　　　　　　　　(**c**)

**Figure 18.** Preparation of specimen: (**a**) specimen bond of cyanoacrylate between rubber and sheet metal; (**b**) specimen in wet cotton and aluminium foil; (**c**) vacuum packed specimen in climatic chamber).

The main emphasis is to validate the derived algebraic model of the torque estimation (design hypothesis) stated a the previous sub-chapter. With rising frequencies and lower temperatures, the affordable drive torque is higher than at slow speeds and warmer conditions. The relation between necessary torque and stiffness (storage modulus), respectively losses (loss modulus), is depending on the selected rubber and should be reproducible for reliable operation of the pump. When the model is sufficient, new generations of such pumps can be designed in one step (simultaneous procedures like material choice, geometry calculations, motor design) without building further prototypes which saves time and money. Additionally, the hydrodynamic fluid gap model (Section 2.1.3) allows to estimate the displacement and flow rate which can also be validated.

The experimental set-up comprised a hydraulic circle with the pump as the main component and included high-, low-pressure lines, a pressure relief valve, a flow meter, pressure as well as temperature sensors, and an on/off valve in the high-pressure line to control the pressure difference. The test rig fits perfectly into a common climatic chamber (450 mm × 450 mm × 450 mm), while the measurement system and the electrical control unit (ECU) controlling the motor were kept outside.

### 2.3.1. Electrical Drive and Performance Determination

The electric commutated brushless direct current motor (EC-BLDC-motor) is controlled by LCMs semi open-source software X2C [28] and powered by the microcontroller unit LCM-ECU-10HB-10A [29] including ten half bridges and some IOs (i.e., analog and digital in- and outputs). The Hall-sensor signals allow a determination of the rotor position, so a state-of-the-art speed controlled, field-oriented control [17] of the BLDC is implemented (see Figure 19). The q-vector-current $i_q$ is proportional to the torque. The motor was embedded into an existing test bench with load and torque measurement.

Starting from zero, the operating points were measured and continuously increased up to 2000 rpm and 35 Watt and the value $i_q$ was evaluated in the motor control unit. The top graph of Figure 20a shows the linear proportionality between torque and $i_q$. The middle graph Figure 20b shows the relation between power, $i_q$ and measured speed, while the efficiency over speed and $i_q$ is illustrated in the bottom diagram Figure 20c. Those values were implemented as look-up tables into the automatic post-processed evaluation script and allow a power and torque determination by measuring $i_q$ and the motor speed. So, for further motor measurements to test the functionality of the pump, neither the applied test bench nor other additional sensors to estimate operation points of electric power, speed and torque is required.

### 2.3.2. Test Rig

The electro-hydraulic scheme is illustrated in Figure 21. The ECU controls the pump speed, and the 2-2-way seat type valve allows digital pressure control (DPC will be explained in the following lines). The climatic chamber heats or cools the test rig to gain insights of required torques for certain speeds, temperatures and loading pressures.

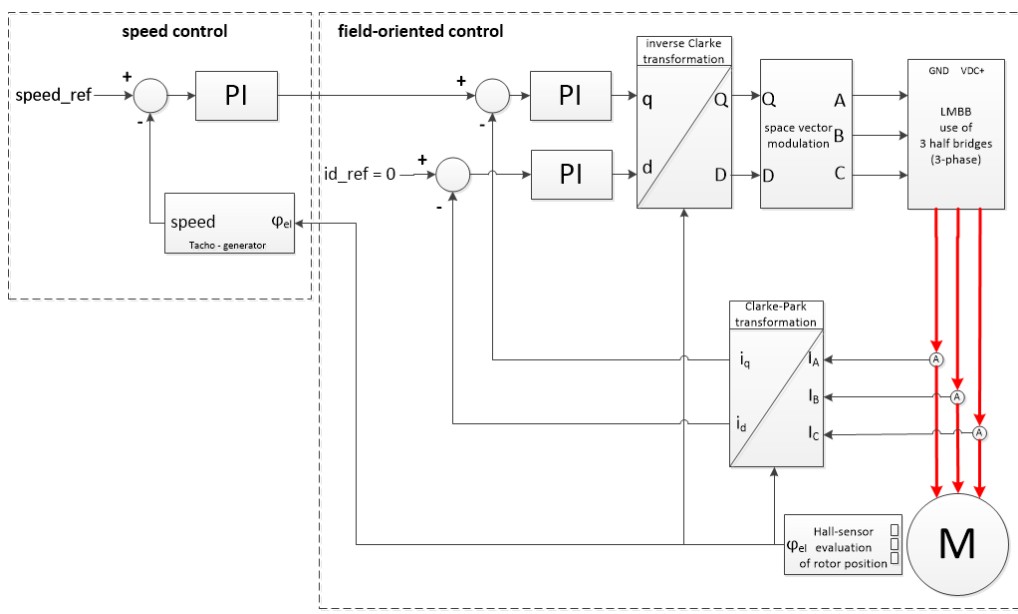

**Figure 19.** Scheme of the speed and field-oriented BLDC motor control.

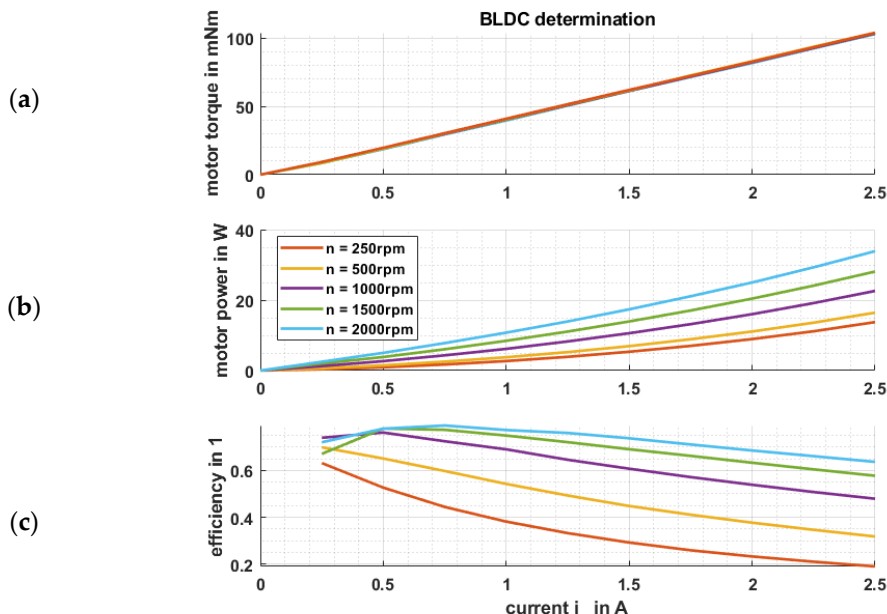

**Figure 20.** BLDC motor performance determination: (**a**) effects of motor speed and q-current variation on motor torque; (**b**) power consumption of the BLDC-motor at certain speeds and loads; (**c**) calculated efficiency graph of the BLDC-motor.

To control a loading pressure on the high-pressure line of the pump, the digital pressure control [30–32] has some benefits in comparison to an ordinary control with a proportional valve. The applied normally closed 2-2-way seat type valve can close the pump outlet without fluid flow (leakage) in order to test the pump in the worst-case operation situation. To keep the pressure at a certain value, the valve will be excited with a duty cycle signal according to pulse width modulation (see Figure 22). A higher duty cycle reduces the mean resistance of the orifice at a certain flow rate; hence, the loading pressure sinks, and vice versa. In combination with an ordinary PI-controller with flow rate depending feed forward, this leads to a smooth pressure loading setting [33].

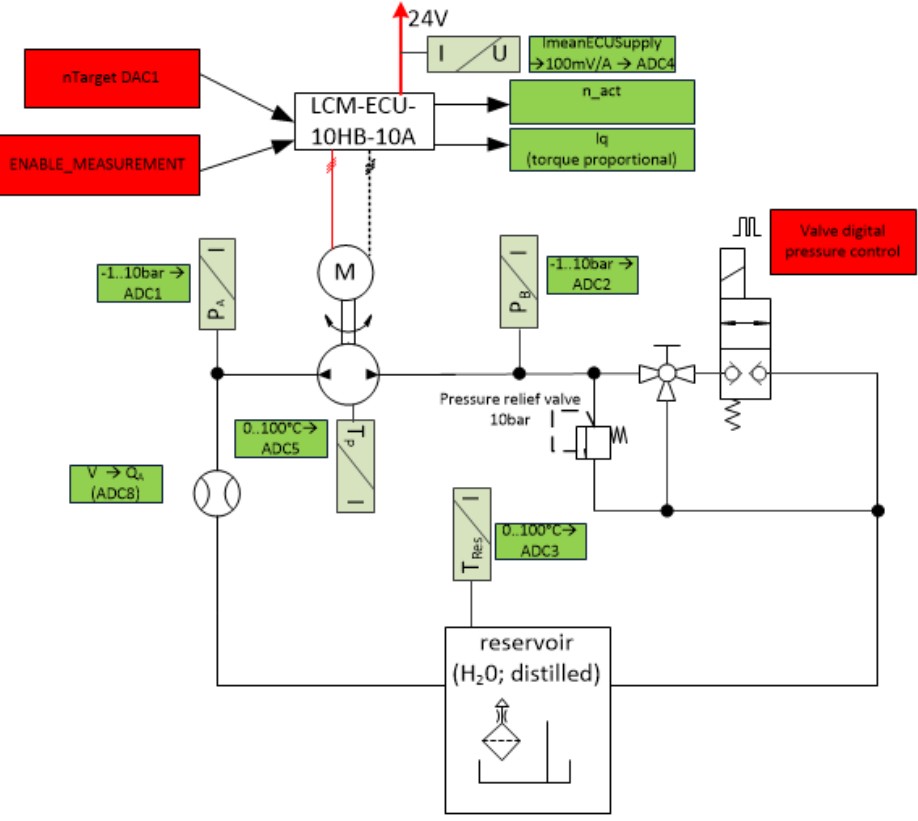

**Figure 21.** Scheme of the test rig (hydraulic and electrical circuit).

### 2.3.3. Definition of the Test Cycle

For the performance tests, an automation script [34] was written to carry out the measuring procedure [35]. The objective was to estimate the pump behaviour within ambient test conditions. The climate chamber temperature range was chosen between 0 °C and up to 40 °C. The loading pressure was set between 0 MPa and 0.5 MPa, and the motor speed ramped up to 1500 rpm. Figure 23 shows a test cycle at 20 °C. The results are post-processed to estimate the displacement and its long-term behaviour as well as the required torque to maintain the pump's displacement at various conditions.

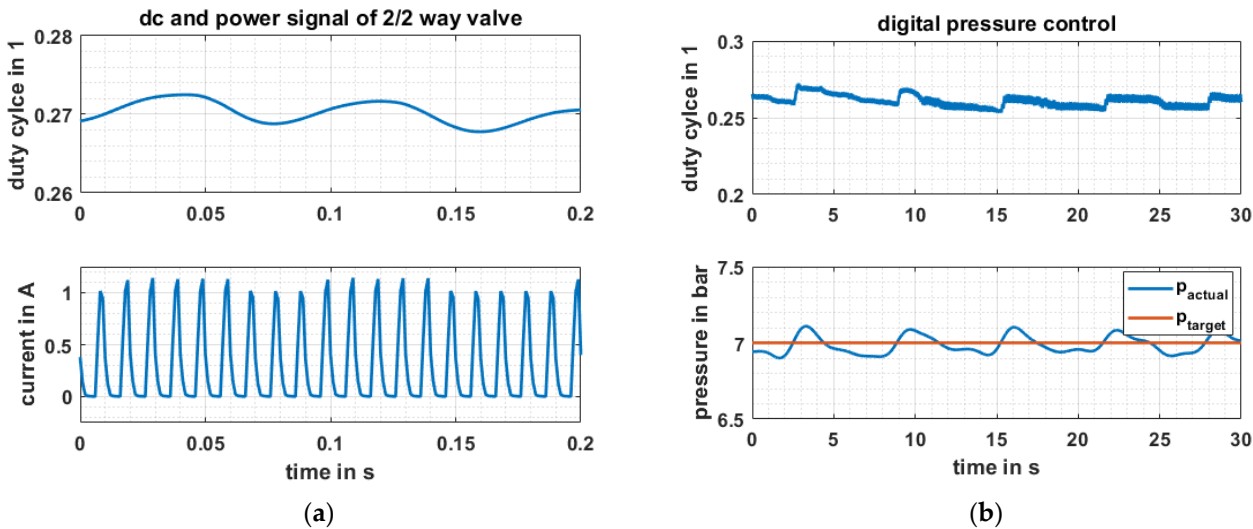

**Figure 22.** The DPC can be used whenever there is a sort of capacity on the pressure side. (**a**) power signal of a boosted 2-2-way seat valve; (**b**) graph of a pressure control including target and actual pressure.

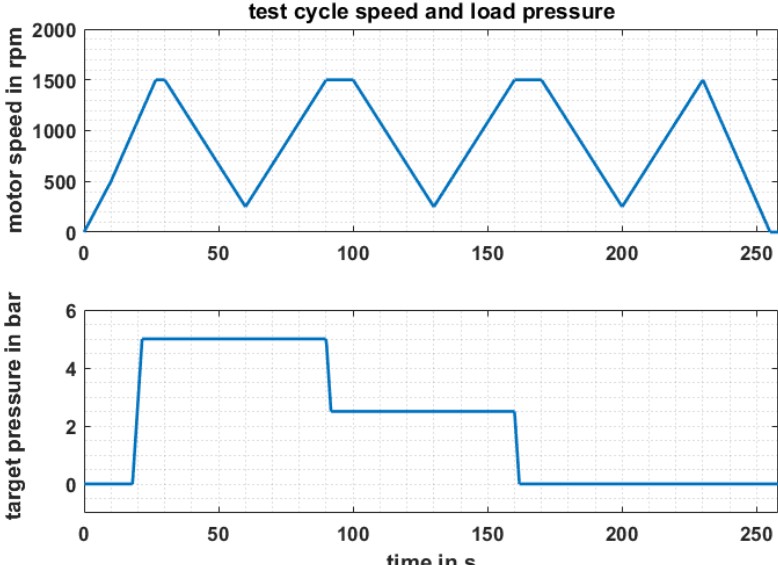

**Figure 23.** Test cycle: motor speed and pressure over time; One v-shaped ramp down and up for each load pressure.

## 3. Results and Discussion

### 3.1. Material Selection of Wobble Plate's Soft Layer

The first criterial to be characterized was the loss modulus E″ and its temperature as well as frequency dependencies. In Figure 24 the E″ (T, f) characteristics of the three candidate materials (TPU, NBR and LSR) are presented. LSR fits the best to the requirements of low E″ within the temperature range of 0 °C and 60 °C. Also the sensitivity to temperature and loading frequency changes is lower compared to the other materials. Another material property requirement for the soft layer of the wobble plate is to be flexible and reveal low, resilient modulus. Figure 25a shows the Mooney-Plot of the candidate materials. The lowest values are measured for LSR and NBR. Taking into account the E″ (T, f) characteristics, the material of choice is LSR with Shore A hardness of 70. Furthermore, the rate dependency of the LSR's hyperelastic material behavior is low confirming the DMA data of Figure 24. The loading rate dependent equi-biaxial characteristics of LSR are analyzed in the Mooney-Plot of Figure 25b. Only a small parallel shifting is observable and, therefore, this formulation is perfectly suitable for the designated application. From these Mooney-Plots the hyperelastic material parameter for the well-known Mooney–Rivlin constitutive model can be easily derived by linear fitting of the reduced stresses and inverted stretches (of higher order for equi-biaxial data). These material parameters are needed for the FE analyses of the soft layer by calculating the contact pressure as well as the resulting stresses (strains) under loading. With these results the design of the soft layer's geometry can be optimized iteratively.

### 3.2. Bonding between the Wobble Plate's Soft and Rigid Layers

As Figure 26a,b reveal, the cataplasm test has shown that the bonding with cyanoacrylate could not resist those extreme conditions. All 3 specimens fall apart during unpacking even without additional mechanical forces. It can be said that cyanoacrylate respective adhesive bonding is no opportunity for this application. Therefore, an injection moulded and vulcanized connection between soft layer with the wobble plate (even with form closure) as mentioned in Section 2.1.4 and shown in Figure 17 should be considered for the prototype.

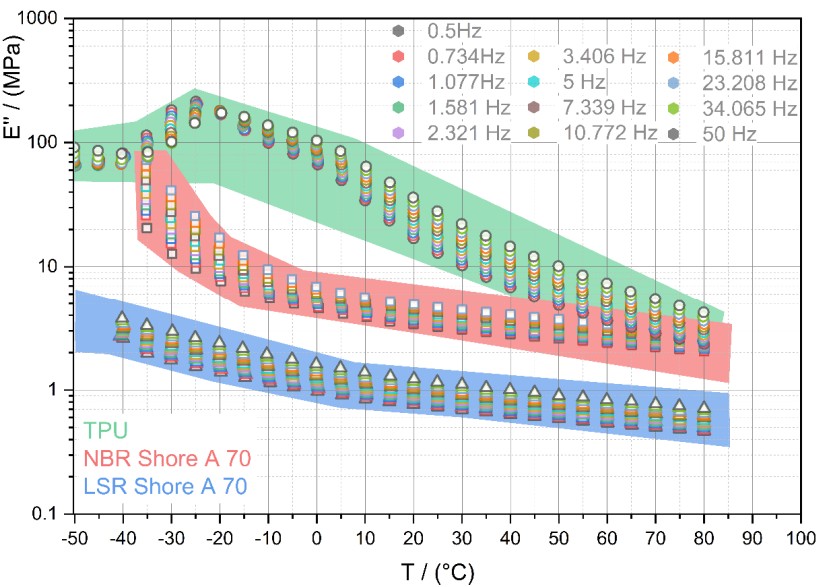

**Figure 24.** Temperature and frequency dependent loss modulus E″ of the candidate materials TPU, NBR and LSR.

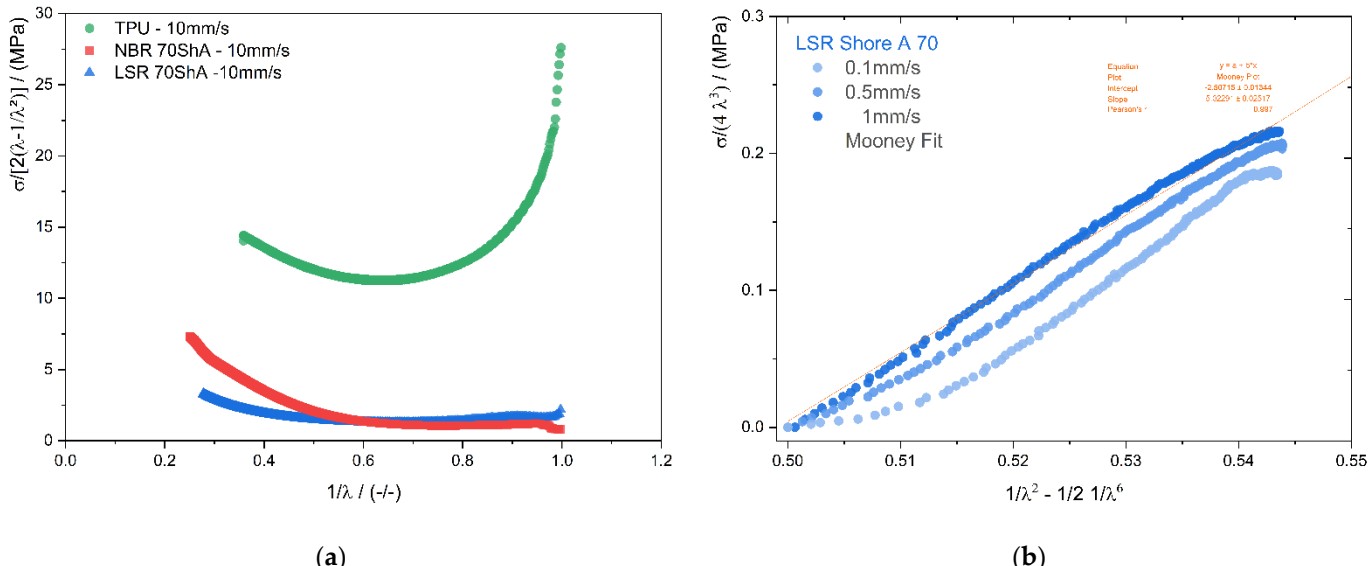

(**a**)                                                                    (**b**)

**Figure 25.** Hyperelastic characteristics evaluated in Mooney Plots under (**a**) uniaxial loading for all candidate materials (TPU, NBR and LSR); (**b**) equi-biaxial loading of the selected material liquid silicone rubber (LSR) with a hardness of Shore A 70.

### 3.3. Determination of the Static Displacement

After assembling the first measurement was the displacement of the pump at room temperature and atmospheric conditions. Tubes (inner diameter 'd' = 4 mm) were connected to the in- and outlet ports of the pump. The system was filled with distilled water and the pump was slowly turned manually n times, which is easily performed since the primary BLDC-drive consists of an external rotor. To produce an accurately measurable amount of pumped water, the pump was turned n times, cumulating n times the pump displacement of several µL.

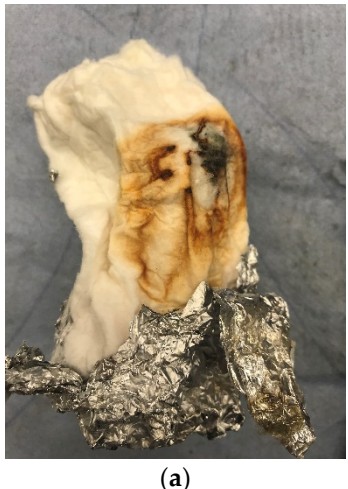
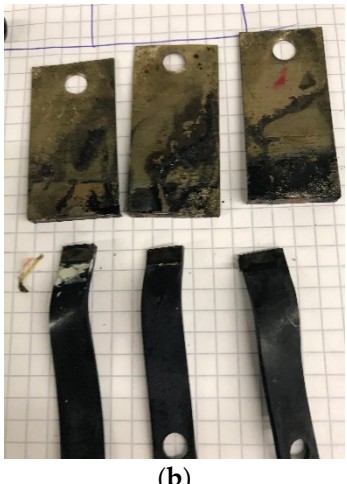

(**a**)    (**b**)

**Figure 26.** Evaluation of the cataplasm-test: (**a**) opening the vacuum-packed specimen; (**b**) comparison of the 3 specimens.

The position difference $\Delta l$ between start and end of motion was measured to calculate quasi-static displacement as followed:

$$V_0 = \frac{\Delta l \cdot d^2 \cdot \pi}{4 \cdot n}, \tag{24}$$

For better correlation of the calculated and the measured fluid displacement, the geometric deviation from the ideal CAD shape is crucial. Figure 27 shows the sealing lip deviation of the CAD contour in comparison to the manufactured component measured with a 3D surface scanner (Keyence 3D Profilometer VR-5000). A deviation in displacement within $\pm 25\%$ is predictable and acceptable. The designed volume was calculated to be about 40 µL. The actual displacement for a quasistatic turn of the pump was measured to 34 µL. In this case, the actual maximum deviation is about 20%. In order to reach the same flow rate as desired, the motor speed must be increased properly.

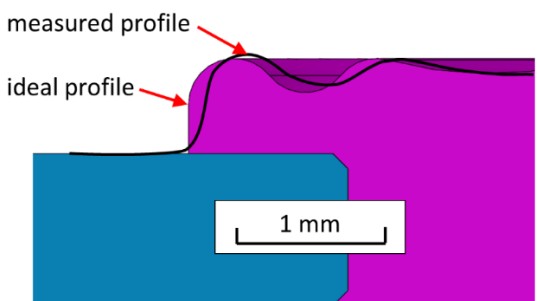

**Figure 27.** Investigation of the manufactured sealing lip; contour comparison of CAD (edge of the pink area) and manufactured rubber (bold black line) measured via white light scanning.

### 3.4. Pump Testing

Besides the controlled parameters and filtered measured values (motor speed, feed pressure and flow rate), the results consist of the post-processed, calculated values, such as drive torque and power generated from the filtered measurements. Figure 28 shows the whole test cycle results for a certain climatic chamber temperature (20 °C). Figure 28a contains the target as well as measured speed and load pressures. In Figure 28b the most important results were illustrated, namely torque and flow rate. The third graph (Figure 28c) shows the evaluated powers. As expected, higher pressures lead to higher necessary drive torques and the flow rate is nearly proportional to the motor speed Figure 29 shows the

same procedure as Figure 28 but for two different climatic chamber conditions 0 °C (a–c) and 40 °C (d–f).

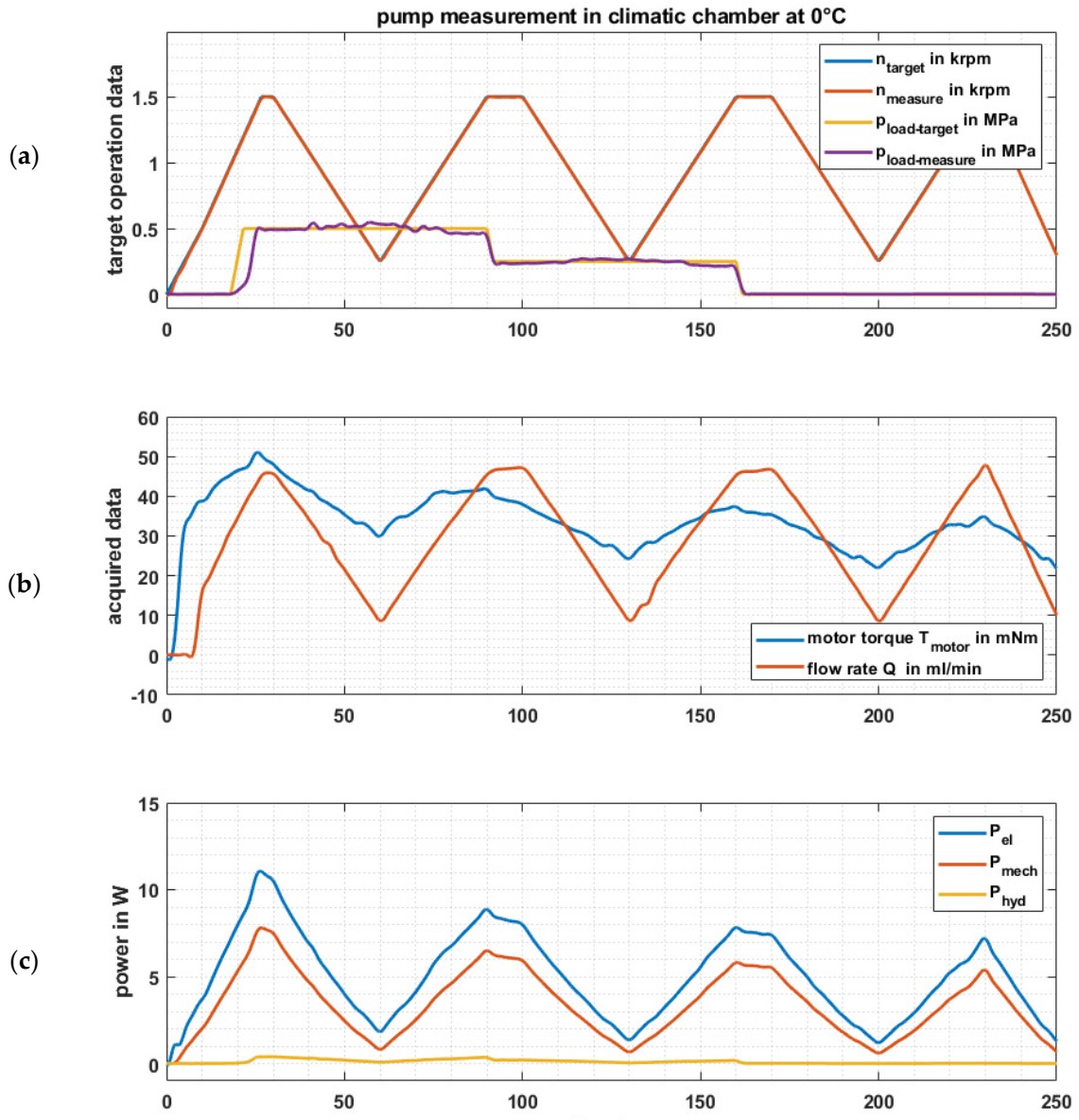

**Figure 28.** Results of the test cycle at room temperature (20 °C); (**a**) target test cycle motor speed n and load pressure p; (**b**) measured torque and flow rate; (**c**) electrical, mechanical and hydraulic power.

Figure 30a shows the measured drive torque characteristic and its temperature as well as motor speed dependency. Here, we can observe that the highest torque is necessary at the coldest temperature, the highest pressure, and the highest motor speed. Losses due to the rubber's entropy elasticity lead to an inner (adiabatic) heating reducing the storage modulus and, thus, the overall loss factor. These losses are not implemented in the theoretical consideration. At elevated temperatures, in this case 40 °C, the required torques reveal low frequency, respectively motor speed, dependency. Figure 30b illustrates the theoretically estimated drive torques for the pump, which can be directly compared to the measured results in Figure 30a. At low motor speeds, the torque estimation is sufficiently accurate, however the absolute value of the torque at low temperatures and high frequencies is rising but significantly lower as estimated. The maximum measured torque of 50 mNm in Figure 29b (at 0 °C and 1500 rpm) exceeds the calculated result

of approximately 40 mNm by 25%. The post-processed results exhibit the maximum at 45 mNm, which were evaluated by averaging the results for both speed ramps (up and down) at same conditions, hence, the inner (adiabatic) heating reduces the mean torque. The edge operation point (maximum motor speed and lowest temperature) is the critical state to dimension for the primary drive. So, the predicted (calculated) values at elevated temperatures are not crucial for the operation in terms of torque requirement. The observed difference of 25% between the prediction and the measurement is within the deviations of the experimental determination of the material data (storage as well as loss moduli). Deviations in the pump's testing and evaluations are superimposed to those of the material characterizations.

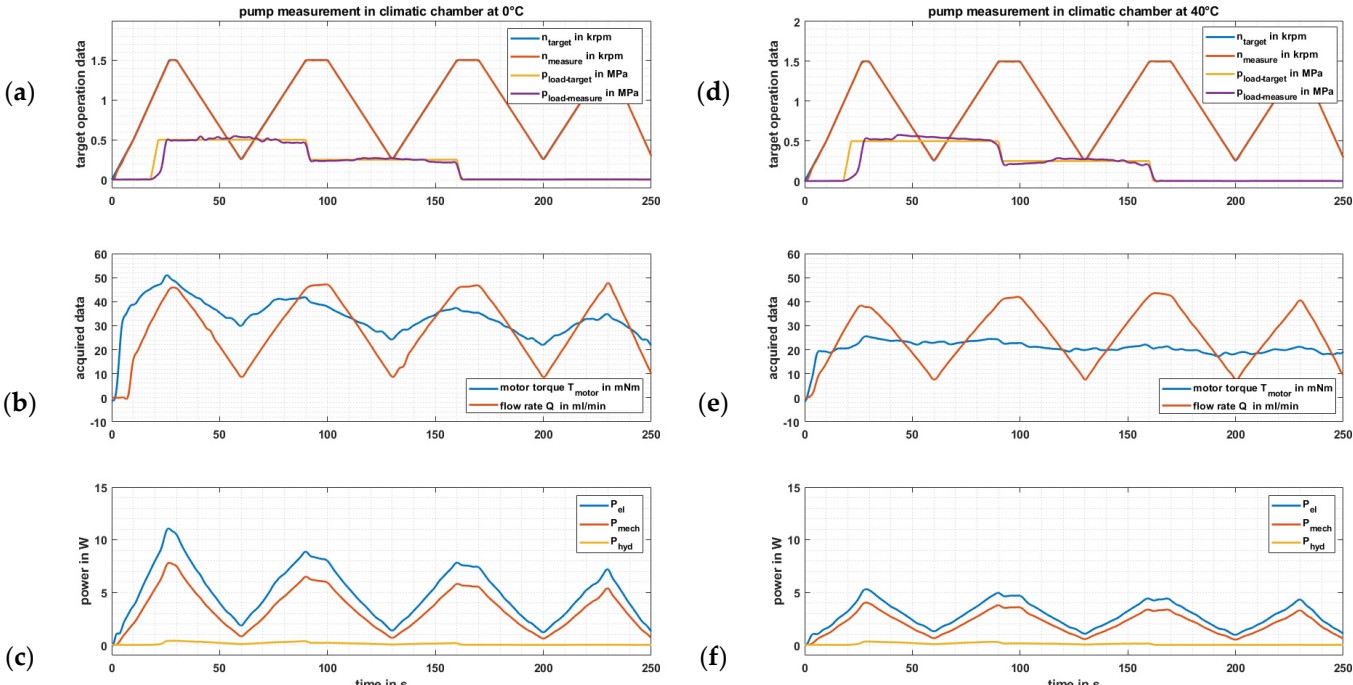

**Figure 29.** Results are structured as in Figure 28 but for different temperatures; (**a**–**c**) results of the test cycle at $0°$; (**d**–**f**) results of the test cycle at $40\,°C$.

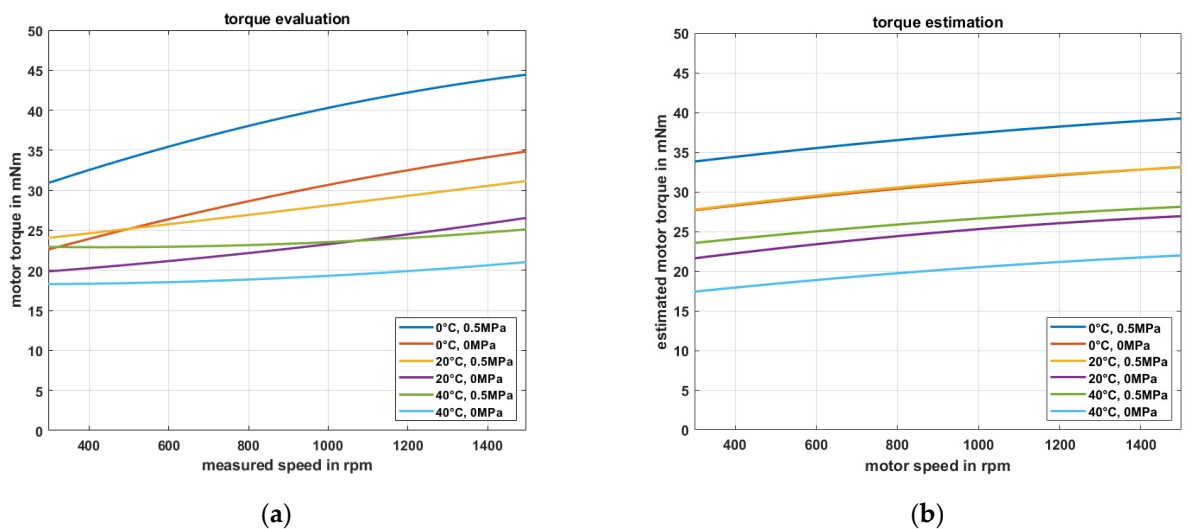

**Figure 30.** Torque validation: (**a**) Illustration of the approached torque curve depending on speed at all measured conditions regarding temperature and load pressure; (**b**) results of torque estimation as stated at Equation (13) for material and geometry data of prototype.

The pump testing and evaluation procedure revealed that the algebraic equations based on empirical material data and geometric parameters are sufficiently reliable to predict the drive torque in the crucial operation points of the pump. The standard deviation of 25% of the pump's displacement is observed in the worst-case operation point and is within the scatter of the conducted experiments and includes the error propagation (noise, environmental as well as signal fluctuations, among others). Therefore, a conservative safety factor of 50% is suggested for the prediction of the target torque and, hence, the selection of the peristaltic pump's primary drive.

The displacement evaluations depending on operation temperature, motor speed (frequency), and loading pressure was calculated as:

$$V(f,T,p) = \frac{Q_{measure}}{n\_measure},$$

(25)

The comparison of the model-based prediction of the pump's displacement to the measured displacement shows (see Figure 31) that a similar decline of the displacement with increasing motor speed is observed at both loading pressures (0 MPa and 0.5 MPa). However, the measured displacement characteristics have a steeper decline. At 1500 rpm the pump's displacement decreases about 10%; the calculated drop by the proposed model is only 2%. The reason for such a deviation in displacement from estimation to measurement is due to the insufficiency of the abstracted model. A parallel plate movement was considered in the model as a simplification, however the (wobbling) plate is rather tilted than parallel. Further dynamic effects such as damping of the rubber, inertia of the wobble plate and inflation of the enclosed fluid volume were not considered and may have more influence on the pump's displacement than assumed. Furthermore, the stiffness of the wobbling plate was set to be linear in contrast to the more complex hyper-elastic rubber behaviour, which was used for the 2D-axissymmetric FE-simulation. Finding the exact displacement was not the aim of this investigation, rather it was to gain insights into its characteristics and, more importantly, ensure steady-state displacement at a specific operation point. So, leakage is prevented meaning that there is no backflow from the high to low pressure side.

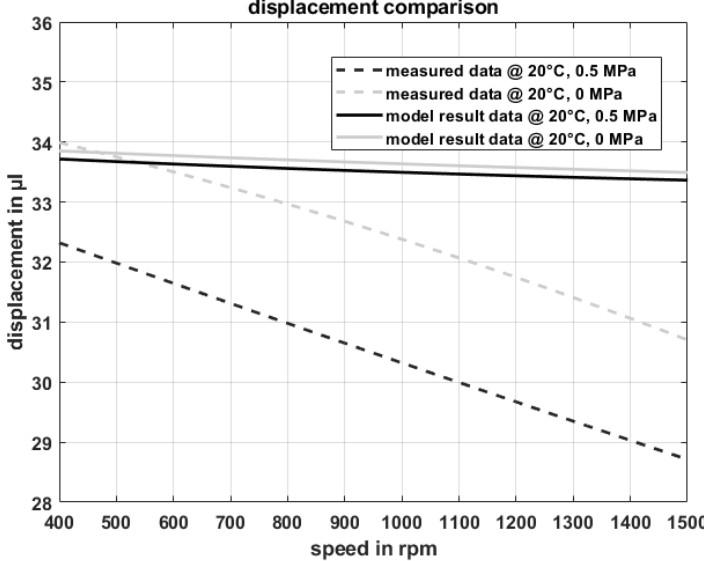

**Figure 31.** Displacement characteristics at 20 °C; idle and nom. load (5 bar) and varying frequency; measurement vs. simple algebraic model (the graph was corrected with a constant factor to reach the measured initial displacement at 0 rpm to equalize the causes due to manufacturing tolerances and assembling).

## 4. Conclusions and Outlook

The peristaltic pump's primary drive torque has a functional relation to boundary conditions, geometry as well as material parameters. This simple model can sufficiently predict the needed torque for new proportional scaled designs by using geometry parameters, safety factor and rubber material data. Therefore, the whole pump including the drive can be designed, manufactured, and verified at once. Our proposed model and design methodology are an alternative to complex and long-lasting costly dynamic 3D FE-simulations. Figure 32 shows a graph to illustrate the designing process.

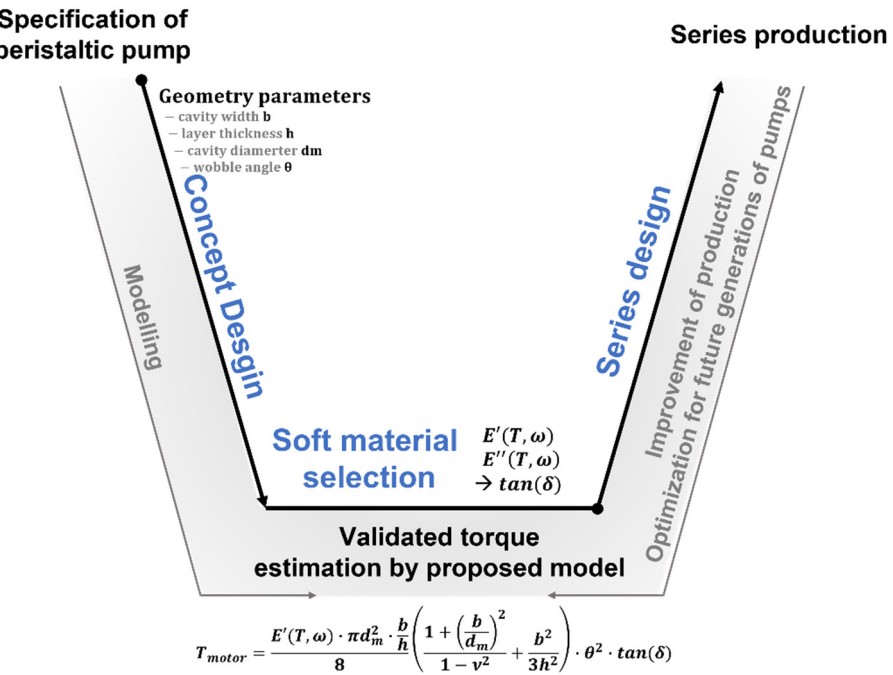

**Figure 32.** V-model of the evaluated design methodology.

With the determined specific values of temperature, frequency and loading dependent drive torque, the applicability of the proposed methodology to design a sufficiently algebraic (quasistatic) physical model for the torque which depends on the rubber material's behaviour (empirical data) as well as boundary and operating conditions can be verified. Determination of the empirical data must be performed with precaution as the rubber behaviour is inherently hyper- and viscoelastic. A high sensitivity to environmental (temperature and humidity) changes can lead to significant alterations of the dynamic thermomechanical properties. Additionally, stress/strain softening effects and hysteretic (adiabatic) heating must be characterized for reliable operation of the peristaltic micro-dosing pump. Therefore, it is of particular importance to formulate (tailor) the rubber's material behaviour to exhibit low loss properties ($E''$ and $\tan(\delta)$) along with high durability (mechanical as well as thermal induced aging).

The peristaltic pump's displacement has pronounced nonlinear motor speed (frequency) and temperature dependencies mainly caused the interaction of the well-known rubber and the dynamic fluid gap (respective fluid cushioning). To get valuable predicted flow rate results, further investigations of the fluid gap model and the interaction with the rubber are necessary. Also, slight changes in the concept design can reduce those deviations. Scheidl et al. [36] addressed this cushioning groove problem and presents a solution by reducing the section of contact to a minimum.

After finding an appropriate pump design and estimating the required torque, the primary drive can be optimized.

**Author Contributions:** Conceptualization, A.P. and T.Z.; methodology, A.P., T.Z. and U.D.Ç.; software, T.Z.; validation, A.P., T.Z., U.D.Ç. and C.E.; formal analysis, T.Z. and U.D.Ç.; investigation, A.P., T.Z., U.D.Ç. and C.E.; resources, A.P., T.Z., U.D.Ç. and C.E.; data curation, A.P., T.Z., U.D.Ç. and C.E.; writing—original draft preparation, T.Z. and U.D.Ç.; writing—review and editing, T.Z. and U.D.Ç.; visualization, A.P., T.Z., U.D.Ç. and C.E.; supervision, A.P.; project administration, A.P. and U.D.Ç. All authors have read and agreed to the published version of the manuscript.

**Funding:** This research received no external funding.

**Institutional Review Board Statement:** Not applicable.

**Informed Consent Statement:** Not applicable.

**Data Availability Statement:** Not applicable.

**Acknowledgments:** Open Access Funding was received by the University of Linz. The presented research work has partly been supported by the Johannes Kepler University, Institute for Polymer Product Engineering (Zoltán Major), the companies Erwin Mach Gummitechnik GmbH, Helmut Hechinger GmbH & CO KG and the Linz Center of Mechatronics GmbH (LCM), which is part of the COMET/K2 program of the Federal Ministry of Transport, Innovation and Technology and the Federal Ministry of Economics and Labor of Austria. The authors would like to thank the Austrian and Upper Austrian Government for their support.

**Conflicts of Interest:** The authors declare no conflict of interest.

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
