# Peer review of "Mechanical Design and Performance Analyses of a Rubber-Based Peristaltic Micro-Dosing Pump"

_actuators, doi:10.3390/act10080198_

Round 1

Reviewer 1 Report

Thank you for the carefull revision. Still, the motivation and application area could be pointed out better, but now the manuscript is very transparent.
Just two recommondations/ideas are left:
- figure 32 could have the same structure as figure 2
- the description of the FE-simulation is very detailed and helpful, but the information about the global size of 0,12mm has no relation without any information of the size of the sealing. Better leave it out, add a scale bar on the simulation picture or give one information about the size of the sealing.

Author Response

Please find our reply in the attached PDF.

Reviewer 2 Report

The article provides quite detailed information about the design and performance of a rubber-based peristaltic micro-dosing pump. The information provided by the authors is reasonable and at a sufficiently high scientific level. The results of experimental research are presented and described in detail. The conclusions are based on the results of research. The proposed model and methodology are valuable and can be used as an alternative for similar studies.

Some notes for authors to improve quality of the article:

- The quality of some Figures could be improved. For example, there is almost no difference between the curves of different colors in Figures 9 (bottom), 22 (b) (bottom), etc.

- The are quite a few recent years articles from scientific journals in the list of references. Meanwhile, the list of references includes a number of publications by the authors themselves.

Author Response

(The authors gave the same response as above.)

Reviewer 3 Report

In this manuscript, to develop a peristaltic mirco-dosing pump, mechanical design, analysis and experimental verification were conducted. The methods and verification results were described definitely and sufficiently.

It is considered to be almost acceptable in present form.

Please check and modify some figures to make the information clearer;

  • Fig. 3(a): the extended figure should be drawn outside.
  • Characters in some figures are unsharp, expecially Figs. 16, 17.

Author Response

Please find our reply in the attached PDF.

This manuscript is a resubmission of an earlier submission. The following is a list of the peer review reports and author responses from that submission.

Round 1

Reviewer 1 Report

Brief summary: The paper presents an innovative pump concept that has a high potential to compete with the widely used peristaltic pumps in the field of microfluidics. Furthermore, an effective design method is proposed for sizing this kind of pump. As the design process is described very in detail and the valdiation is analysed very critical, this paper provides a contribution of high importance. It reveals the possibility to use a very simple approach to estimate a pump characteristics including a highly nonlinear material, namely an elastomer as well as the limitation of the approach and the importance of a detailed description of the viscoelastic material behavior.

Broad comments: As mentioned in the brief summary, this paper presents an innovative pump concept and an effective approach for the design by combination of FE-simulation and analytical approaches. Unfortunately this is not well expressed in the abstract and introduction. The adressed scope of application is not well defined and therefor the motivation of the specified requirements is not transparent. For example, micropumps are widely used in the fields of medical and laboratory technology. In these application areas a pressure difference of under 0,5 MPa is usual.  
A very extensive research was done including theoretical analysis as well as a high effort in experiments to adress lots of aspects regarding the most important influences on the pumps performance (e.g. material behavior, joint of the sealing and aging). To improve the transparency, the illustrations should be improved by more details (e.g. exact contur of the sealing in fig. 3). It is very difficult to understand how the theoretical approaches, models and experimental data are connected. This could be improved either with an addittional scheme or by adding the solution approach to figure 2 for each step, the use of a consistent naming in the text and the assignment of this model names to figure 32. 

Reviewer 2 Report

Article is too descriptive in well known parts ale not well organised, abstract is not a summary of article, in introduction instead of general introduction to the problem is already own design of authors..really not well written. Most of the references are in german language and I am pretty sure, that there are english equivalents for basic equations of hydrodynamic (Navier-Stokes or Reynolds equations). Whole text is so long and descriptive, that I totally miss the novelty of presented design, is there any? What was the purpose of the article? According to my opinion the article needs too many editions, needs to be rewritten in less details about known methods and constructionas, it must be clearly pointed out what is new and compare it with the commonly used desings. Also the references must be mostly english.

Some more technical notes, there are some equations with no nomenclature or description of equation terms. Some of well known eqautions are written in generally not really common shape (Reynolds and Navier-Stokes) with reffernce to german literature and no terms descriptions. Article abstract consist of some numbers which is higly unusual and does not summarize the article output same as the conclusion, that does not show the impact of the article for the researcher group of possible readers.